# Self-Improved Prior for All-in-One Image Restoration

## Abstract

Unified image restoration models for diverse and mixed degradations often suffer from unstable optimization dynamics and inter-task conflicts. This paper introduces Self-Improved Privilege Learning (SIPL), a novel paradigm that overcomes these limitations by innovatively extending the utility of privileged information (PI) beyond training into the inference stage. Unlike conventional Privilege Learning, where ground-truth-derived guidance is typically discarded after training, SIPL empowers the model to leverage its own preliminary outputs as *pseudo-privileged signals* for iterative self-refinement at test time. Central to SIPL is Proxy Fusion, a lightweight module incorporating a learnable Privileged Dictionary. During training, this dictionary distills essential high-frequency and structural priors from privileged feature representations. Critically, at inference, the *same* learned dictionary then interacts with features derived from the model's initial restoration, facilitating a self-correction loop. SIPL can be seamlessly integrated into various backbone architectures, offering substantial performance improvements with minimal computational overhead. Extensive experiments demonstrate that SIPL significantly advances the state-of-the-art on diverse all-in-one image restoration benchmarks. For instance, when integrated with the PromptIR model, SIPL achieves remarkable PSNR improvements of +4.58 dB on composite degradation tasks and +1.28 dB on diverse five-task benchmarks, underscoring its effectiveness and broad applicability.

## 1 Introduction

All-in-one image restoration, which aims to tackle diverse and often mixed degradations with a single, unified model, has emerged as a pivotal research area due to its immense practical value Jiang et al. (2025). However, these versatile models confront a fundamental dilemma: forcing a single network to master the distinct, often confusing or conflicting, feature representations required for tasks like denoising (local textures) and dehazing (global context) inevitably leads to optimization challenges and performance compromises Kong et al. (2024); Wu et al. (2024). This heterogeneity brings unstable training process, resulting in the suboptimal solution. To address this challenge, we argue for a paradigm shift against the end-to-end learning. We propose to explicitly learn and leverage a universal prior of what constitutes a high-quality, degradation-free image, using it as a stable guide to navigate the complex optimization landscape.

To realize this, we draw inspiration from Privilege Learning (PL) Vapnik and Vashist (2009); Vapnik et al. (2015), where auxiliary information is used to guide training progressively while can be unavailable at test time. We posit that ground-truth (GT) images can serve as the ultimate privileged information, providing a clear, degradation-agnostic supervisory signal. The incorporation of privileged information during training establishes an *inter-task comprehension bridge*, effectively harmonizing conflicting gradients and stabilizing convergence progressively. For instance, a PL-enhanced PromptIR model achieves performance comparable to the original baseline in less than half the training epochs (see Figure 1(a)), showcasing that PL therefore presents an efficient and straightforward strategy for improving existing all-in-one restoration methods. Nevertheless, vanilla PL traditionally limits its own impact, as the privileged guidance is often progressively diminished during training and is entirely unavailable at inference time.

This raises a critical question: *can the essence of privileged information be retained and repurposed to enhance performance at the inference stage?* We answer in the affirmative by proposing the novel

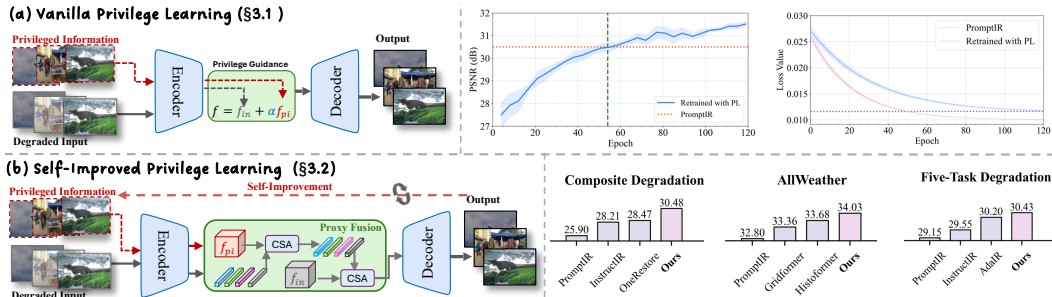

Figure 1: Conceptual comparison of learning frameworks: (a) Privilege Learning (PL) leverages privileged information during training for improved optimization. (b) Our proposed Self-Improved Privilege Learning (SIPL) framework introduces Proxy Fusion to retain privileged knowledge, enabling iterative self-refinement during inference by using intermediate restoration outputs as pseudo-privileged information. Retrained PromptIR with the proposed SIPL achieves significant improvement across diverse all-in-one tasks.

**Self-Improved Privilege Learning (SIPL)** framework, which extends the conventional PL paradigm in two key stages, as illustrated in Figure 1(b). First, we introduce a the Privileged Dictionary (PD) to **distill and retain universal high-quality priors** from privileged information during training. Second, we propose an iterative self-refinement cycle to **reuse these retained priors** at inference, where the model's own initial restoration serves as *pseudo-privileged information* for a powerful self-refinement loop.

In detail, this two-stage concept is realized by our proposed *Proxy Fusion* mechanism, a straightforward and efficient module centered around a learnable **Privileged Dictionary (PD)**. During training, the PD internalizes degradation-agnostic image characteristics from GT-derived features, creating a compact and persistent knowledge base. At inference, the PD's role transforms from a student to a teacher. It interacts with features extracted from the model's own preliminary restoration (the pseudo-privileged information). This interaction allows the immutable, pre-trained priors within the PD to guide a feature-level correction, progressively enhancing the output in a self-driven manner. The efficacy of this approach is evident when integrating SIPL with the PromptIR model, which yields remarkable PSNR gains of **+4.58 dB** on Composite Degradation, **+1.23 dB** on Allweather, and **+1.38 dB** on the Five-Task benchmark, as illustrated in Figure 1(b).

Our contributions can be summarized into four aspects:

- We introduce Privilege Learning (PL) to all-in-one image restoration, achieving a stronger and more stable optimization baseline by effectively mitigating inter-task conflicts.

- We extend the conventional PL framework with a novel mechanism to **retain** privileged knowledge for the inference stage, thus overcoming the ephemeral nature of guidance.

- We further propose a self-refinement strategy to **reuse** this retained knowledge. This empowers models with a new capability for iterative self-improvement, leading to promising performance boosts.

- We propose the **Proxy Fusion**, a lightweight and plug-and-play module that makes SIPL a practical reality. Its high efficiency and broad compatibility are validated by significant performance gains on multiple state-of-the-art architectures.

## 2 RELATED WORK

### 2.1 ALL-IN-ONE IMAGE RESTORATION

Traditional image restoration typically targets specific degradations, such as noise or blur, using specialized models Zamir et al. (2022); Huang et al. (2023); Cai et al. (2023). However, real-world scenarios often involve unknown or mixed degradations, driving the need for *all-in-one* models that handle diverse degradation types in a unified framework Su et al. (2022-05); Jiang et al. (2025).

Early efforts toward unified models utilized powerful backbones Chen et al. (2022a; 2021); Wang et al. (2022); Zamir et al. (2022); Wang et al. (2024); Guo et al. (2024a) or generative approaches like diffusion models Belhasin et al. (2024); Yue and Loy (2024); Li et al. (2025); Yue et al. (2025). A key challenge is guiding a single network to handle diverse restoration tasks, leading to a focus

on conditioning mechanisms, from early degradation encoders Li et al. (2022) to advanced learnable prompts Potlapalli et al. (2023), sometimes refined with frequency priors or dimensionality reduction Cui et al. (2025); Zhang et al. (2023). Other methods use degradation priors from classifiers Kong et al. (2024) or vision-language models like CLIP with textual prompts Luo et al. (2024); Lin et al. (2024). Additionally, multi-task learning techniques Wu et al. (2024; 2025) and pretraining strategies Qin et al. (2024) address optimization challenges. While these advances improve all-in-one capabilities, they often focus on architectural or input conditioning changes, overlooking optimization stability in multi-task learning.

## 2.2 PRIVILEGE LEARNING AND TEST-TIME ADAPTATION

**Privilege Learning (PL)** introduced the concept of using extra, training-only information to help a model learn a more robust representation Vapnik and Vashist (2009); Vapnik et al. (2015). While explored in low-level vision Lee et al. (2020), its application has been limited. We first establish PL for all-in-one restoration to achieve a more stable training baseline. More importantly, we extend PL to *retain* and *reuse* this privileged knowledge at inference, overcoming its core limitation.

**Test-Time Adaptation (TTA)** aims to adapt pre-trained models to new test domains via online, unsupervised fine-tuning Liang et al. (2025). Although our self-refinement also operates at test time, it is fundamentally different. TTA *updates model parameters* using gradient-based optimization on test data to address domain shift Deng et al. (2023); Gou et al. (2024); Yang et al. (2024). In stark contrast, SIPL is a *gradient-free inference process* that operates on a *frozen model*. It does not adapt the model but instead refines the output by reusing pre-distilled knowledge, making it a distinct paradigm from TTA.

## 3 METHOD

In this section, we detail our Self-Improved Privilege Learning (SIPL) framework. We first provide an overview of how SIPL integrates into a general restoration architecture. We then elaborate on its two core stages: (1) the PL-enhanced training process that retains privileged knowledge, and (2) the inference-time interation that reuses this knowledge for iterative self-refinement.

## 3.1 PRELIMINARIES

**Privilege Learning Paradigm** As established in Section 1, all-in-one image restoration faces fundamental optimization challenges stemming from task competition and conflicting gradient directions. Privilege Learning Vapnik and Vashist (2009); Vapnik et al. (2015) offers an elegant solution by leveraging additional information during training that enhances the learning process. Given a degraded image $I_d$ and its corresponding ground truth $I_{gt}$, PL incorporates privileged information (derived from $I_{gt}$) into the training process. This is achieved through a simple yet effective feature fusion approach:

$$F_{fused} = (1 - \alpha) \cdot F_d + \alpha \cdot F_{PI}, \tag{1}$$

where $F_d$ represents features extracted from the degraded image, $F_{PI}$ denotes features derived from the ground truth (privileged information), and $\alpha \in [0, 1]$ controls the degree of privileged guidance. During training, $\alpha$ typically follows a decreasing schedule, ensuring the model gradually adapts to operating without privileged guidance. At inference time, $\alpha$ is set to 0, as privileged information is unneeded.

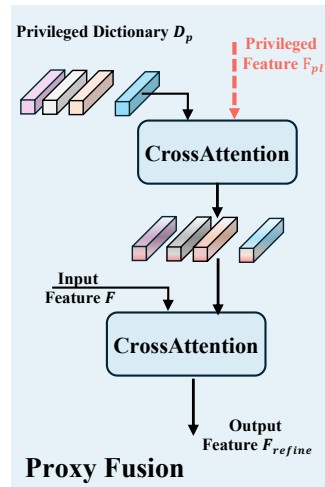

Figure 2: Implementation of the proxy fusion module.

This straightforward mechanism provides substantial benefits for all-in-one restoration. It stabilizes training and mitigates task competition by providing clear guidance, particularly for challenging degradation types. Meanwhile, our introduced PL paradigm remains agnostic to model architecture, enhancing all-in-one image restoration generally.

## 3.2 SELF-IMPROVED PRIVILEGE LEARNING

Beyond the vanilla PL paradigm, we propose a novel extension to retain the privileged information at the inference stage. Our key insight is that the restoration output, though imperfect, contains valuable information closer to the ground truth than the original

degraded input. This observation motivates our proxy fusion module, which enables the model to leverage its own outputs as pseudo-privileged information during inference.

**Proxy Fusion** The cornerstone of SIPL is our novel Proxy Fusion mechanism, which creates a persistent bridge between training-time privileged knowledge and inference-time self-improvement. Unlike the direct feature blending in conventional PL (Eq. 1), Proxy Fusion employs a learnable Privileged Dictionary (PD) to distill and retain essential knowledge from privileged information:

$$\text{PD} \in \mathbb{R}^{N \times C}, \tag{2}$$

where $N$ represents the number of dictionary entries and $C$ is the feature dimension. This dictionary interacts with privileged features through cross-attention:

$$F'_{PI} = \text{CrossAttention}(Q = \text{PD}, K = F_{PI}, V = F_{PI}), \tag{3}$$

This interaction allows the PD to capture and internalize high-frequency details and statistical patterns characteristic of high-quality images. The distilled knowledge is then integrated with features from the degraded image:

$$F_{fused} = \text{CrossAttention}(Q = F_d, K = F'_{PI}, V = F'_{PI}). \tag{4}$$

During training, the entire framework, including the Privileged Dictionary, learns to extract and utilize privileged information effectively. Crucially, the learned PD persists into inference, serving as a knowledge repository that enables self-improvement without requiring actual ground truth.

**Self-Improvement Mechanism** The distinguishing feature of SIPL is its iterative self-refinement capability during inference:

1. **Initial Restoration**: The model produces an initial output $I^{(0)}_{restored}$ using only the degraded input:

$$I^{(0)}_{restored} = \mathcal{F}(I_d). \tag{5}$$

2. **Self-Improvement**: This initial output serves as pseudo-privileged information. Features extracted from $I^{(0)}_{restored}$ interact with the learned PD through the Proxy Fusion mechanism, guiding subsequent restoration:

$$I^{(1)}_{restored} = \mathcal{F}(I_d, I^{(0)}_{restored}). \tag{6}$$

3. **Iterative Refinement (Optional)**: This process can continue for multiple iterations, with each step potentially improving quality:

$$I^{(t)}_{restored} = \mathcal{F}(I_d, I^{(t-1)}_{restored}), \quad t \geq 1. \tag{7}$$

This elegant self-correction loop enables progressive quality enhancement without requiring actual ground truth during deployment. In practice, we find that a single refinement step ($t = 1$) typically provides substantial improvements with minimal computational overhead.

The key advantage of our Proxy Fusion approach over direct feature blending is its ability to distill and retain essential high-quality image characteristics in the learned PD parameters. This creates a persistent knowledge repository that facilitates self-improvement during inference, a capability absent in conventional PL frameworks.

### 3.3 REMARKS

Our SIPL framework focuses exclusively on the learning paradigm rather than specific architectural choices. The Proxy Fusion module serves as a plug-and-play component that can enhance virtually any existing restoration architecture. This architectural agnosticism ensures broad applicability across diverse restoration models and tasks. For experimental validation, we integrate SIPL into multiple distinct backbone architectures, including PromptIR Potlapalli et al. (2023). As demonstrated in Section 4, SIPL consistently improves performance across all tested models, confirming its generality and effectiveness as an optimization framework for all-in-one image restoration. For a detailed theoretical analysis of the optimization dynamics and stability guarantees underpinning our framework, please refer to Appendix A.

Table 1: Quantitative results (PSNR/SSIM) on the Three-Task Setting. Our results are highlighted in **bold**, and best results are underlined.

| Type | Method | Venue | Denoising (BSD68) | | | Dehazing | Deraining | Average |
|---|---|---|---|---|---|---|---|---|
| | | | $\sigma = 15$ | $\sigma = 25$ | $\sigma = 50$ | SOTS | Rain100L | |
| *General* | MPRNet Zamir et al. (2021) | CVPR'21 | 33.27/0.920 | 30.76/0.871 | 27.29/0.761 | 28.00/0.958 | 33.86/0.958 | 30.63/0.894 |
| | Restormer Zamir et al. (2022) | CVPR'22 | 33.72/0.930 | 30.67/0.865 | 27.63/0.792 | 27.78/0.958 | 33.78/0.958 | 30.75/0.901 |
| | NAFNet Chen et al. (2022a) | ECCV'22 | 33.03/0.918 | 30.47/0.865 | 27.12/0.754 | 24.11/0.928 | 33.64/0.956 | 29.67/0.844 |
| | FSNet* Cui et al. (2022) | TPAMI'23 | 33.81/0.930 | 30.84/0.872 | 27.69/0.792 | 29.14/0.968 | 35.61/0.969 | 31.42/0.906 |
| | DRSformer* Chen et al. (2023) | CVPR'23 | 33.28/0.921 | 30.55/0.862 | 27.58/0.786 | 29.02/0.968 | 35.89/0.970 | 31.26/0.902 |
| | MambaIR* Guo et al. (2024a) | ECCV'24 | 33.88/0.931 | 30.95/0.874 | 27.74/0.793 | 29.57/0.970 | 35.42/0.969 | 31.51/0.907 |
| *All-in-One* | DL Fan et al. (2019) | TPAMI'19 | 33.05/0.914 | 30.41/0.861 | 26.90/0.740 | 26.92/0.391 | 32.62/0.931 | 29.98/0.875 |
| | AirNet Li et al. (2022) | CVPR'22 | 33.92/0.932 | 31.26/0.888 | 28.00/0.797 | 27.94/0.962 | 34.90/0.967 | 31.20/0.910 |
| | IDR* Zhang et al. (2023) | CVPR'23 | 33.89/0.931 | 31.32/0.884 | 28.04/0.798 | 29.87/0.970 | 36.03/0.971 | 31.83/0.911 |
| | Gridformer* Wang et al. (2024) | IJCV'24 | 33.93/0.931 | 31.37/0.887 | 28.11/0.801 | 30.37/0.970 | 37.15/0.972 | 32.19/0.912 |
| | NDR Yao et al. (2024) | TIP'24 | 34.01/0.932 | 31.36/0.887 | 28.10/0.798 | 28.64/0.962 | 35.42/0.969 | 31.51/0.910 |
| | InstructIR Conde et al. (2024) | ECCV'24 | 34.15/0.933 | 31.52/0.890 | 28.30/0.804 | 30.22/0.959 | 37.98/0.978 | 32.43/0.913 |
| | TextualDegRemoval Lin et al. (2024) | CVPR'24 | 34.01/0.933 | 31.39/0.890 | 28.18/0.802 | 31.63/0.980 | 37.58/0.979 | 32.63/0.917 |
| | AdaIR Cui et al. (2025) | ICLR'25 | 34.12/0.935 | 31.45/0.892 | 28.19/0.802 | 31.06/0.980 | 38.64/0.983 | 32.69/0.918 |
| | PromptIR Potlapalli et al. (2023) | NeurIPS'23 | 33.98/0.933 | 31.31/0.888 | 28.06/0.799 | 30.58/0.974 | 36.37/0.972 | 32.06/0.913 |
| | **PromptIR + SIPL** | 2025 | **34.12/0.933** | **31.48/0.889** | **28.22/0.800** | **31.09/0.977** | **38.43/0.984** | **32.67/0.917** |

Table 2: Quantitative results (PSNR/SSIM) on the Five-Task Setting. Our results are highlighted in **bold**, and best results are underlined.

| Type | Method | Venue | Denoising | Dehazing | Deraining | Deblurring | Low-light | Average |
|---|---|---|---|---|---|---|---|---|
| | | | BSD68 | SOTS | Rain100L | GoPro | LOL | |
| *General* | SwinIR Liang et al. (2021) | ICCVW'21 | 30.59/0.868 | 21.50/0.891 | 30.78/0.923 | 24.52/0.773 | 17.81/0.723 | 25.04/0.835 |
| | Restormer Zamir et al. (2022) | CVPR'22 | 31.49/0.884 | 24.09/0.927 | 34.81/0.962 | 27.22/0.829 | 20.41/0.806 | 27.60/0.881 |
| | NAFNet Chen et al. (2022a) | ECCV'22 | 31.02/0.883 | 25.23/0.939 | 35.56/0.967 | 26.53/0.808 | 20.49/0.809 | 27.76/0.881 |
| | DRSformer* Chen et al. (2023) | CVPR'23 | 30.97/0.881 | 24.66/0.931 | 33.45/0.953 | 25.56/0.780 | 21.77/0.821 | 27.28/0.873 |
| | Retinexformer* Cai et al. (2023) | ICCV'23 | 30.84/0.880 | 24.81/0.933 | 32.68/0.940 | 25.09/0.779 | 22.76/0.834 | 27.24/0.873 |
| | FSNet* Cui et al. (2024) | TPAMI'23 | 31.33/0.883 | 25.53/0.943 | 36.07/0.968 | 28.32/0.869 | 22.29/0.829 | 28.71/0.898 |
| | MambaIR* Guo et al. (2024a) | ECCV'24 | 31.41/0.884 | 25.81/0.944 | 36.55/0.971 | 28.61/0.875 | 22.49/0.832 | 28.97/0.901 |
| *All-in-One* | DL Fan et al. (2019) | TPAMI'19 | 23.09/0.745 | 20.54/0.826 | 21.96/0.762 | 19.86/0.672 | 19.83/0.712 | 21.05/0.743 |
| | TAPE Liu et al. (2022) | ECCV'22 | 30.18/0.855 | 22.16/0.861 | 29.67/0.904 | 24.47/0.763 | 18.97/0.621 | 25.09/0.801 |
| | Transweather Valanarasu et al. (2022) | CVPR'22 | 29.00/0.841 | 21.32/0.885 | 29.43/0.905 | 25.12/0.757 | 21.21/0.792 | 25.22/0.836 |
| | AirNet Li et al. (2022) | CVPR'22 | 30.91/0.882 | 21.04/0.884 | 32.98/0.951 | 24.35/0.781 | 18.18/0.735 | 25.49/0.846 |
| | IDR Zhang et al. (2023) | CVPR'23 | 31.60/0.887 | 25.24/0.943 | 35.63/0.965 | 27.87/0.846 | 21.34/0.826 | 28.34/0.893 |
| | Gridformer* Wang et al. (2024) | IJCV'24 | 31.45/0.885 | 26.79/0.951 | 36.61/0.971 | 29.22/0.884 | 22.59/0.831 | 29.33/0.904 |
| | InstructIR Conde et al. (2024) | ECCV'24 | 31.40/0.887 | 27.10/0.956 | 36.84/0.973 | 29.40/0.886 | 23.00/0.836 | 29.55/0.907 |
| | AdaIR Cui et al. (2025) | ICLR'25 | 31.35/0.889 | 30.53/0.978 | 38.02/0.981 | 28.12/0.858 | 23.00/0.845 | 30.20/0.910 |
| | PromptIR* Potlapalli et al. (2023) | NeurIPS'23 | 31.47/0.886 | 26.54/0.949 | 36.37/0.970 | 28.71/0.881 | 22.68/0.832 | 29.15/0.904 |
| | **PromptIR + SIPL** | 2025 | **31.45/0.888** | **30.51/0.975** | **38.09/0.982** | **29.35/0.886** | **23.23/0.856** | **30.53/0.917** |

# 4 EXPERIMENTS

In this section, we evaluate our SIPL framework on diverse benchmarks (multi-task, deweathering, composite degradation). All models were retrained from scratch following original protocols for fairness. Detailed setups are in the Appendix.

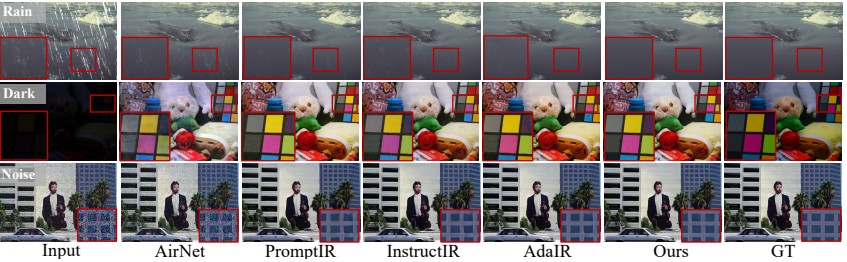

Input    AirNet    PromptIR    InstructIR    AdaIR    Ours    GT

Figure 3: Visual comparison on the Five-Task benchmark. Our method demonstrates superior restoration quality across diverse degradations, effectively recovering finer details and image structures compared to other approaches.

## 4.1 MAIN RESULTS

We now present the quantitative comparison of our SIPL framework, integrated with the PromptIR backbone (denoted as "PromptIR + SIPL"), against the original PromptIR and other state-of-the-art methods across the four benchmark settings.

Table 3: Quantitative results (PSNR/SSIM) on the Deweathering Setting. Our results are highlighted in **bold**, and best results are underlined.

| Method | Venue | Snow100K-S | | Snow100K-L | | Outdoor-Rain | | RainDrop | | Average | |
|---|---|---|---|---|---|---|---|---|---|---|---|
| | | PSNR | SSIM | PSNR | SSIM | PSNR | SSIM | PSNR | SSIM | PSNR | SSIM |
| All-in-One Li et al. (2020) | CVPR'20 | – | – | 28.33 | 0.8820 | 24.71 | 0.8980 | 31.12 | 0.9268 | 28.05 | 0.9023 |
| Transweather Valanarasu et al. (2022) | CVPR'22 | 32.51 | 0.9341 | 29.31 | 0.8879 | 28.83 | 0.9000 | 30.17 | 0.9157 | 30.20 | 0.9094 |
| WGWSNet Zhu et al. (2023) | CVPR'22 | 34.31 | 0.9460 | 30.16 | 0.9007 | 29.32 | 0.9207 | 32.38 | 0.9378 | 31.54 | 0.9263 |
| WeatherDiff$_{64}$ Özdenizci and Legenstein (2023) | TPAMI'23 | 35.83 | 0.9566 | 30.09 | 0.9041 | 29.64 | 0.9312 | 30.71 | 0.9312 | 31.57 | 0.9308 |
| WeatherDiff$_{128}$ Özdenizci and Legenstein (2023) | TPAMI'23 | 35.02 | 0.9516 | 29.58 | 0.8941 | 29.72 | 0.9216 | 29.66 | 0.9225 | 31.00 | 0.9225 |
| AWRCP Ye et al. (2023) | ICCV'23 | 36.92 | 0.9652 | 31.92 | 0.9341 | 31.39 | 0.9329 | 31.93 | 0.9314 | 33.04 | 0.9409 |
| GridFormer Wang et al. (2024) | IJCV'24 | 37.46 | 0.9640 | 31.71 | 0.9231 | 31.87 | 0.9335 | 32.39 | 0.9362 | 33.36 | 0.9392 |
| MPerceiver Ai et al. (2024) | CVPR'24 | 36.23 | 0.9571 | 31.02 | 0.9164 | 31.25 | 0.9246 | 33.21 | 0.9294 | 32.93 | 0.9319 |
| DTPM Ye et al. (2024) | CVPR'24 | 37.01 | 0.9663 | 30.92 | 0.9174 | 30.99 | 0.9340 | 32.72 | 0.9440 | 32.91 | 0.9404 |
| Histoformer Sun et al. (2024) | ECCV'24 | 37.41 | 0.9656 | 32.16 | 0.9261 | 32.08 | 0.9389 | 33.06 | 0.9441 | 33.68 | 0.9437 |
| PromptIR Potlapalli et al. (2023) | NeurIPS'23 | 36.88 | 0.9643 | 31.34 | 0.9200 | 30.80 | 0.9229 | 32.20 | 0.9359 | 32.80 | 0.9357 |
| **PromptIR + SIPL** | 2025 | **37.91** | **0.9673** | **32.34** | **0.9291** | **32.91** | **0.9469** | **32.99** | **0.9462** | **34.03** | **0.9473** |

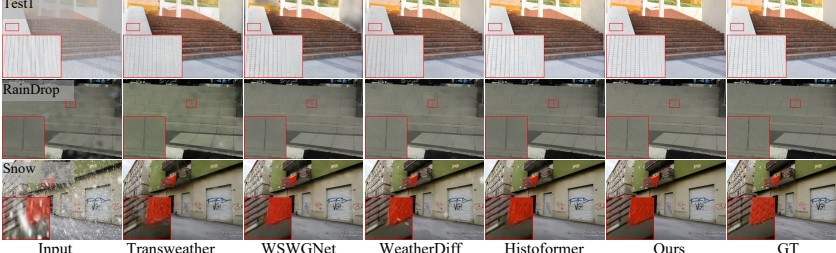

Figure 4: Qualitative examples from the AllWeather dataset. Our method exhibits robust performance in removing various challenging weather conditions. It yields visually superior results with better detail preservation and fewer artifacts.

**Results on Three-Task Setting** As detailed in Table 1, integrating our SIPL framework with the PromptIR backbone yields significant performance gains. SIPL boosts the average PSNR by a notable **+0.61 dB** over the baseline. The benefits are particularly pronounced in challenging tasks like deraining, where SIPL achieves a substantial **+2.06 dB** improvement, and in dehazing (+0.51 dB). Our SIPL-enhanced model is highly competitive against recent state-of-the-art methods, including TextualDegRemoval Lin et al. (2024) and AdaIR Cui et al. (2025).

**Results on Five-Task Setting** On the more demanding five-task benchmark, the advantages of our SIPL framework become even more apparent (Table 2). SIPL significantly elevates the PromptIR baseline, boosting the average PSNR by a substantial **+1.38 dB**. The improvements are particularly striking in tasks where the baseline struggles due to task competition, such as dehazing (**+3.97 dB**) and deraining (**+1.72 dB**), alongside a solid gain in low-light enhancement (+0.55 dB). The visual results in Figure 3 corroborate these findings, showing superior detail and texture preservation across a range of degradations.

**Results on Deweathering Setting** The deweathering benchmark, summarized in Table 3, evaluates performance on removing diverse adverse weather conditions. Our approach again demonstrates superior capabilities, achieving the new SOTA average PSNR of 34.03 dB and SSIM of 0.9473. Consistent performance enhancements are recorded across all four test datasets. These results underscore SIPL's robustness in complex deweathering scenarios, outperforming specialized methods and recent deweathering models like Histoformer Sun et al. (2024) and GridFormer Wang et al. (2024). Qualitative results in Figure 4 corroborate these metrics, showing that our method more effectively removes severe weather artifacts while better preserving fine details and color fidelity.

**Results on Real-World Deweathering Setting** To critically assess the robustness and generalization capability of our framework in complex real-world environments, we conducted an extensive evaluation on the WeatherBench dataset Guan et al. (2025). As detailed in Table 4, SIPL achieves state-of-the-art performance with an average PSNR of **30.66 dB** and SSIM of **0.862**, significantly outperforming the PromptIR baseline by **+2.80 dB**. Notably, our unified model surpasses specialized SOTA methods, beating WGWS-Net in deraining (+0.33 dB) and MWFormer in dehazing (+0.12 dB). These substantial gains confirm that the Privileged Dictionary effectively bridges the synthetic-to-real domain gap. By leveraging degradation-agnostic structural priors learned during training, SIPL guides robust restoration even on unseen real-world artifacts. Visual comparisons in Figure 5 further demonstrate SIPL's superior ability to remove heavy rain streaks and dense haze while preserving structural fidelity and color consistency.

Table 4: Quantitative comparison on the real-world **WeatherBench** dataset. Our method achieves state-of-the-art performance, demonstrating robust generalization across diverse real-world weather conditions. The best and second-best results are highlighted in **bold** and underlined, respectively.

| Method | Venue | Dehaze | | | Derain | | | Desnow | | | Average | | |
|---|---|---|---|---|---|---|---|---|---|---|---|---|---|
| | | PSNR↑ | SSIM↑ | LPIPS↓ | PSNR↑ | SSIM↑ | LPIPS↓ | PSNR↑ | SSIM↑ | LPIPS↓ | PSNR↑ | SSIM↑ | LPIPS↓ |
| DehazeFormer Song et al. (2023) | TIP'23 | 24.12 | 0.745 | 0.345 | 36.05 | 0.954 | 0.181 | 28.88 | 0.849 | 0.178 | 29.68 | 0.849 | 0.235 |
| DCMPNet Zhang et al. (2024) | CVPR'24 | 21.18 | 0.506 | 0.491 | 32.04 | 0.876 | 0.282 | 24.81 | 0.614 | 0.546 | 26.01 | 0.665 | 0.440 |
| DRSformer Chen et al. (2023) | CVPR'23 | 19.95 | 0.694 | 0.404 | 33.98 | 0.943 | 0.209 | 28.00 | 0.836 | 0.197 | 27.31 | 0.824 | 0.270 |
| NeRD-Rain Chen et al. (2024) | CVPR'24 | 21.52 | 0.718 | 0.386 | 35.74 | 0.950 | 0.182 | 28.87 | 0.851 | 0.182 | 28.71 | 0.840 | 0.250 |
| SnowFormer Chen et al. (2022b) | arXiv'22 | 22.71 | 0.736 | 0.305 | 35.18 | 0.951 | 0.155 | 29.30 | 0.868 | 0.143 | 29.06 | 0.852 | 0.201 |
| MPRNet Zamir et al. (2021) | CVPR'21 | 23.27 | 0.739 | 0.355 | 36.14 | 0.954 | 0.171 | 29.18 | 0.860 | 0.177 | 29.53 | 0.851 | 0.234 |
| Restormer Zamir et al. (2022) | CVPR'22 | 19.30 | 0.687 | 0.412 | 34.49 | 0.945 | 0.197 | 27.95 | 0.836 | 0.197 | 27.25 | 0.823 | 0.269 |
| AirNet Li et al. (2022) | CVPR'22 | 20.94 | 0.705 | 0.383 | 33.59 | 0.942 | 0.224 | 22.06 | 0.780 | 0.291 | 25.53 | 0.809 | 0.299 |
| TransWeather Valanarasu et al. (2022) | CVPR'22 | 19.79 | 0.680 | 0.397 | 29.34 | 0.903 | 0.294 | 24.96 | 0.796 | 0.231 | 24.70 | 0.793 | 0.307 |
| WGWS-Net Zhu et al. (2023) | CVPR'23 | 13.79 | 0.603 | 0.535 | 37.08 | **0.961** | **0.117** | 20.81 | 0.780 | 0.248 | 23.89 | 0.781 | 0.300 |
| DiffUIR Zheng et al. (2024) | CVPR'24 | 22.74 | 0.744 | 0.355 | 35.93 | 0.955 | 0.172 | 29.50 | 0.870 | 0.162 | 29.39 | 0.856 | 0.230 |
| MWFormer Zhu et al. (2024) | TIP'24 | 24.42 | 0.746 | **0.284** | 35.15 | 0.951 | 0.153 | 29.98 | **0.872** | **0.133** | 29.85 | 0.856 | **0.190** |
| Histoformer Sun et al. (2024) | ECCV'24 | 17.69 | 0.669 | 0.437 | 30.70 | 0.916 | 0.279 | 25.39 | 0.808 | 0.225 | 24.59 | 0.798 | 0.314 |
| AdaIR Cui et al. (2025) | ICLR'25 | 23.08 | 0.731 | 0.351 | 34.87 | 0.946 | 0.192 | 28.44 | 0.837 | 0.179 | 28.80 | 0.838 | 0.240 |
| PromptIR Potlapalli et al. (2023) | NeurIPS'23 | 21.11 | 0.713 | 0.375 | 34.54 | 0.944 | 0.198 | 27.93 | 0.836 | 0.195 | 27.86 | 0.831 | 0.256 |
| **PromptIR +SIPL** | 2025 | **24.54** | **0.758** | 0.334 | **37.41** | 0.959 | 0.162 | **30.03** | 0.869 | 0.171 | **30.66** | **0.862** | 0.222 |

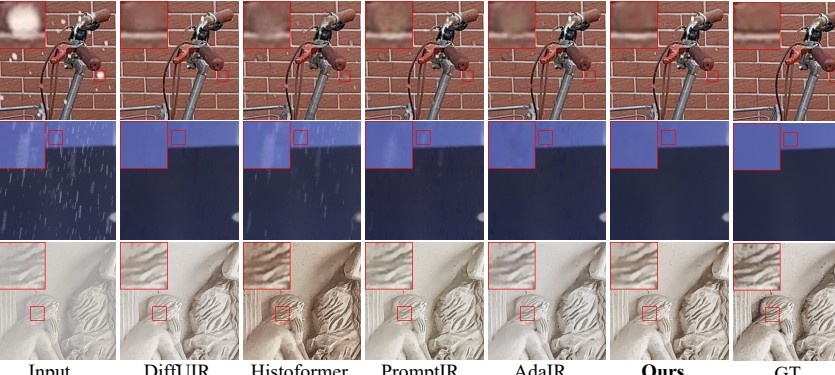

Figure 5: Visual comparisons on the real-world WeatherBench dataset. Our method effectively removes complex weather artifacts while preserving structural details.

**Results on Composite Degradation Setting** The advantages of SIPL are most pronounced on the challenging Composite Degradation benchmark (Table 5), where multiple degradations interact. Here, our approach achieves a remarkable **+4.58 dB** average PSNR improvement over the PromptIR baseline. This substantial gain, far exceeding that of PL-enhanced training alone, underscores the power of our inference-time self-refinement. The performance leap is particularly notable on severe combined degradations like haze+snow (**+9.01 dB**) and haze+rain (**+8.16 dB**). As shown in Figure 6, unlike baseline methods that typically address only one degradation, SIPL demonstrates superior generalization, effectively mitigating all co-occurring artifacts and restoring details and color fidelity.

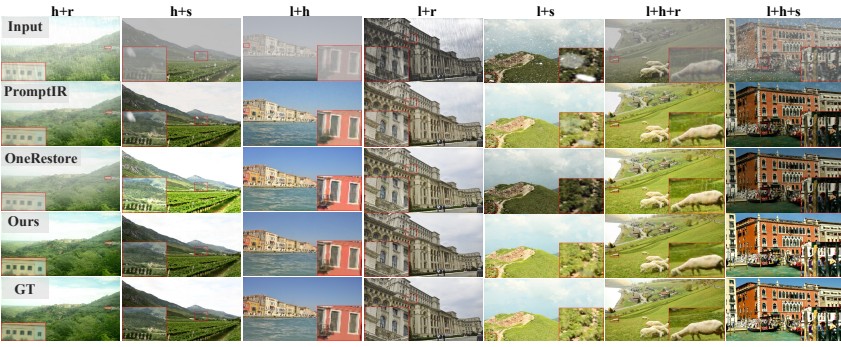

Figure 6: Visual results on the composite degradation tasks, showcasing performance on mixed degradations. Our method more effectively mitigates multiple interacting degradations, restoring clearer images with improved color fidelity and detail.

Table 5: Quantitative results (PSNR/SSIM/LPIPS) on the Composite Degradation Setting. Our results are highlighted in **bold**, and best results are underlined.

| Method | Venue | l | h | r | s | l+h | l+r | l+s | h+r | h+s | l+h+r | l+h+s | Avg. |
|---|---|---|---|---|---|---|---|---|---|---|---|---|---|
| AirNetLi et al. (2022) | CVPR'22 | 24.83 | 24.21 | 26.55 | 26.79 | 23.23 | 22.82 | 23.29 | 22.21 | 23.29 | 21.80 | 22.24 | 23.75 |
| TransWeatherValanarasu et al. (2022) | CVPR'22 | 23.39 | 23.95 | 26.69 | 25.74 | 22.24 | 22.62 | 21.80 | 23.10 | 22.34 | 21.55 | 21.01 | 23.13 |
| WeatherDiffÖzdenizci and Legenstein (2023) | TPAMI'23 | 23.58 | 21.99 | 24.85 | 24.80 | 21.83 | 22.69 | 22.12 | 21.25 | 21.99 | 21.23 | 21.04 | 22.49 |
| WGWSNetZhu et al. (2023) | CVPR'23 | 24.39 | 27.90 | 33.15 | 34.43 | 24.27 | 25.06 | 24.60 | 27.23 | 27.65 | 23.90 | 23.97 | 26.96 |
| InstructIRConde et al. (2024) | ECCV'24 | 26.70 | 32.61 | 33.51 | 34.45 | 24.36 | 25.41 | 25.63 | 28.80 | 29.64 | 24.84 | 24.32 | 28.21 |
| OneRestoreGuo et al. (2024b) | ECCV'24 | 26.55 | 32.71 | 33.48 | 34.50 | 26.15 | 25.83 | 25.56 | 30.27 | 30.46 | 25.18 | 25.28 | 28.47 |
| PromptIRPotlapalli et al. (2023) | NeurIPS'23 | 26.32 | 26.10 | 31.56 | 31.53 | 24.49 | 25.05 | 24.51 | 24.54 | 23.70 | 23.74 | 23.33 | 25.90 |
| **PromptIR + SIPL** | 2025 | **27.62** | **36.82** | **35.66** | **36.85** | **27.03** | **26.79** | **26.68** | **32.70** | **32.71** | **26.20** | **26.20** | **30.48** |

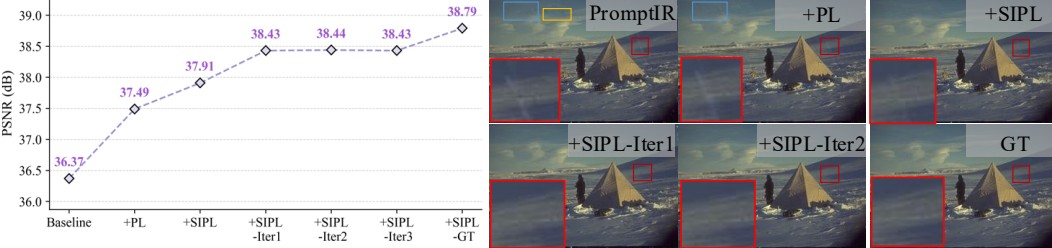

Figure 8: Ablation study dissecting the contributions of SIPL's components. The figure illustrates the progressive improvements from the baseline, through the addition of privilege learning ("+PL"), the initial application of SIPL ("+SIPL"), and subsequent iterative self-refinement stages ("+SIPL-IterX"). Performance is benchmarked against an approximate upper bound using GT-guidance ("+SIPL-GT").

## 4.2 ABLATION STUDIES

In this section, we conduct comprehensive ablation studies to meticulously validate the efficacy of our proposed SIPL framework and dissect the contributions of its core components.

**Architectural Agnosticism of SIPL** A core strength of our SIPL framework lies in its architectural agnosticism and ease of integration. To substantiate this plug-and-play capability, we applied SIPL to a spectrum of distinct backbone architectures, moving beyond the PromptIR Potlapalli et al. (2023). These included Restormer Zamir et al. (2022), a prominent Transformer-based network; NAFNet Chen et al. (2022a), known for its high CNN efficiency; and AdaIR Cui et al. (2025), a recent state-of-the-art method notable for its frequency domain processing. All models were retrained on the five-task benchmark with and without the SIPL

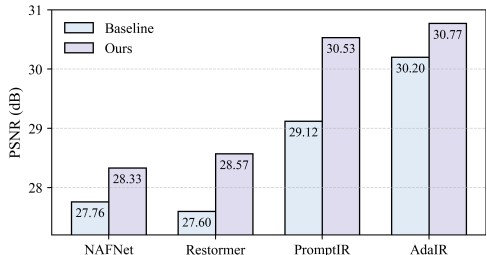

Figure 7: Ablation study on the interaction of SIPL with diverse backbone architectures on the five-task benchmark.

framework integrated. The results, presented in Figure 7, unequivocally showcase SIPL's ability to consistently elevate performance across these diverse architectural paradigms. More detailed results are available at Appendix.

**Dissecting the Contributions of SIPL Components** To meticulously evaluate SIPL's core components, we ablate PromptIR on a five degradation tasks, with quantitative and qualitative results presented in Figure 8. The baseline PromptIR model achieves a PSNR of 36.37 dB. Introducing privilege learning solely during training ("+PL") substantially boosts performance to 37.49 dB. This underscores PL's efficacy in stabilizing multi-task optimization by leveraging privileged information, thereby establishing a stronger foundation. Building upon this, the full SIPL framework, by incorporating the Proxy Fusion module for its initial inference application ("+SIPL"), further elevates the PSNR to 37.91 dB (+0.42 dB over "+PL"). This increment highlights the Proxy Fusion module's critical role, with its Privileged Dictionary, in effectively distilling, preserving, and transferring privileged knowledge for tangible improvement at inference time. The iterative self-improvement mechanism, a key innovation of SIPL, demonstrates further significant refinement. Crucially, when the model's output from the "+SIPL" stage is *first fed back as pseudo-privileged information* (resulting in "+SIPL-Iter1"), performance impressively surges to 38.43 dB—an additional +0.52 dB gain.

This substantial improvement from the initial feedback loop powerfully demonstrates the innovation and effectiveness of using self-generated outputs for refinement. While subsequent iterations show diminishing returns, they affirm the model's capacity for self-correction. This iterative process using pseudo-PI effectively narrows the gap towards the performance upper bound of 38.79 dB, achieved when utilizing true ground truth as privileged input during inference.

**Impact of Privileged Dictionary Size.** To determine the optimal capacity for the Proxy Fusion module, we analyze the effect of the Privileged Dictionary (PD) size $N$ on the five-task restoration performance. As illustrated in Figure 9, there is a clear positive correlation between the dictionary size and model performance. Specifically, increasing $N$ from 32 to 256 yields steady improvements, validating that a larger dictionary can distill and retain more diverse high-frequency structural priors from the ground-truth supervision. Notably, expanding the dictionary size from the initial $N = 256$ to $N = 512$ results in a significant performance leap, boosting the average PSNR from 30.53 dB to 30.79 dB. However, further scaling to $N = 1024$ offers only marginal gains (+0.03 dB), indicating that the representation capability has reached saturation. Considering the trade-off between restoration accuracy and the computational overhead associated with larger attention maps, we adopt $N = 512$ as the default setting for our final model. Detailed numerical results are provided in the Appendix.

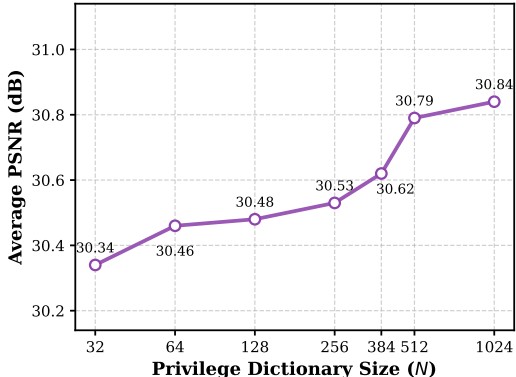

Figure 9: Ablation study on the Privileged Dictionary size ($N$). The curve illustrates the average PSNR on the Five-Task benchmark as $N$ increases. Performance improves significantly up to $N = 512$ and then saturates.

Table 6: Average PSNR (dB) across ten self-refinement iterations on the Three-Task and Five-Task benchmarks. The most significant performance gain occurs at the first iteration ($t = 1$), after which the performance rapidly converges to a stable plateau, demonstrating the robustness and stability of our iterative refinement mechanism. Detailed per-task metrics are available in the appendix.

| Benchmark | Baseline | t=1 | t=2 | t=3 | t=4 | t=5 | t=6 | t=7 | t=8 | t=9 | t=10 |
|---|---|---|---|---|---|---|---|---|---|---|---|
| Three-Task | 32.06 | 32.669 | 32.670 | 32.673 | **32.673** | **32.673** | 32.671 | 32.670 | 32.669 | 32.669 | 32.669 |
| Five-Task | 29.15 | 30.53 | 30.58 | 30.58 | **30.60** | 30.58 | 30.55 | 30.53 | 30.54 | 30.52 | 30.53 |

**Evaluation on Multi-Step Refinement** To comprehensively analyze the behavior and stability of our iterative self-refinement, we evaluated its performance over ten successive steps. Table 6 presents the average PSNR progression on both the Three-Task and Five-Task benchmarks, with peak performance highlighted in bold. The results reveal two crucial insights into our framework's dynamics. First, the most substantial performance leap universally occurs at the initial refinement step ($t = 1$). On the Three-Task benchmark, this single step yields a gain of +0.609 dB, while on the more complex Five-Task benchmark, the improvement is an even more remarkable +1.38 dB. This demonstrates that our mechanism can immediately and effectively correct the most significant errors in the initial restoration. Second, beyond this initial jump, the performance rapidly saturates and converges to a stable plateau. Subsequent iterations from $t = 2$ to $t = 10$ result in only marginal fluctuations (e.g., within $\pm0.004$ dB on the Three-Task benchmark), with performance remaining consistently high and peaking around the fourth iteration. This graceful convergence is a hallmark of our method's robustness. It stands in stark contrast to naive feedback loops, which, as our prior ablation showed, lead to catastrophic performance collapse due to error amplification. The stability of SIPL is attributed to the Privileged Dictionary (PD), which acts as a constant, reliable guide, ensuring that the feature-level corrections do not diverge. Based on this analysis, our choice of a single refinement step ($t = 1$) for the main experiments is a principled one, offering an optimal balance between substantial performance enhancement and computational efficiency.

**Analysis of the Self-Improvement Mechanism** To rigorously validate superiority of our proposed self-improvement mechanism, we compare it against two common inference-time enhancement strategies: naive iteration and self-ensembling. As shown in Table 7, simply feeding a model's output back as its input (Naive Iteration) results in a catastrophic performance collapse of -3.82 dB.

Table 7: Ablation study on different inference-time enhancement strategies on the Three-Task benchmark. Our proposed SIPL framework not only significantly outperforms the baselines in restoration quality (PSNR/SSIM) but also demonstrates superior computational efficiency compared to standard self-ensembling.

| Method | PSNR (Avg.) | SSIM (Avg.) | Forward Passes | Mechanism |
|---|---|---|---|---|
| Baseline (PromptIR) | 32.06 | 0.913 | 1× | Standard Inference |
| + Naive Iteration | 28.24 (-3.82) | 0.892 | 2× | Pixel-level Reprocessing |
| + Self-Ensembling | 32.42 (+0.36) | 0.915 | 8× | Averaging Augmented Outputs |
| **+ SIPL (Ours)** | **32.67 (+0.61)** | **0.917** | **2×** | **PD-Guided Feature Refinement** |

This failure is anticipated: the initially restored image, while visually improved, constitutes an out-of-distribution (OOD) input for a model trained on heavily degraded data. This domain mismatch leads to the amplification of residual artifacts and color shifts, causing the model to diverge. This experiment unequivocally demonstrates that effective self-correction is a non-trivial task that cannot be achieved by simple looping. Unlike naive iteration's pixel-level reprocessing, SIPL performs a principled **feature-level refinement**. The initial output's features are not used to generate a new image directly, but rather to query the pre-learned Privileged Dictionary (PD). This PD, having distilled the essence of high-quality images during training, acts as a robust guardrail, providing a stable, gradient-free correction signal in the latent space.

Moreover, self-ensembling, which averages the outputs from eight geometrically augmented inputs, provides a moderate performance gain (+0.36 dB). However, our SIPL framework surpasses it with a more substantial improvement of +0.61 dB. Notably, SIPL achieves this superior result with only 2 forward passes (for one refinement step as the default), making it approximately **4× more computationally efficient** than the 8-pass self-ensembling process. The stability of this process is further corroborated by our multi-iteration experiments (as shown in Table 7), where performance gracefully saturates near the GT-guided upper bound rather than collapsing. This highlights the robustness and fundamental advantage of our PD-guided self-improvement paradigm. Further analyses are detailed in the Appendix.

### 4.3 LIMITATION AND FUTURE WORK

While our experiments have demonstrated the significant success and versatility of the SIPL framework, we identify several limitations and promising avenues for future research. First, while we empirically validate that Privilege Learning (PL) stabilizes multi-degradation training, a deeper theoretical understanding of its optimization dynamics in this context remains an open question. A more formal understanding in this area is a critical challenge for the community, which could unlock even more profound improvements. Second, our iterative refinement, while more efficient than ensembling, still increases latency over single-pass baselines; optimizing this performance-cost trade-off is a key priority. Looking forward, SIPL's task- and model-agnostic design makes it a promising candidate for a wider range of restoration problems, especially in complex real-world scenarios. More broadly, our self-refinement concept resonates with the reasoning capabilities of recent large models. Future work could explore integrating principles from paradigms like Chain-of-Thought Wei et al. (2022), potentially using large vision models to guide more sophisticated, multi-step correction processes and unlock new capabilities in low-level vision.

### 5 CONCLUSION

In this work, we propose Self-Improved Privilege Learning (SIPL), a novel framework that effectively tackles critical optimization impediments in all-in-one image restoration. SIPL uniquely extends the paradigm of privilege learning to the inference stage: models are empowered to iteratively self-refine their outputs by leveraging them as pseudo-privileged information. This is realized through our proposed proxy fusion module, which employs a privileged dictionary, distilled from ground-truth priors during training, to guide this self-correction process with retrained privileged prior. Extensive evaluations across multiple challenging benchmarks, particularly those with complex composite degradations, confirm that SIPL substantially boosts the performance of diverse state-of-the-art methods, significantly enhancing their robustness and overall restoration quality. We hope that the principles and methodologies presented in SIPL will offer fresh perspectives to the community and stimulate further exploration into more effective strategies for all-in-one image restoration.

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

## A THEORETICAL ANALYSIS AND REMARKS

To provide a rigorous justification for the empirical success of SIPL, we formally analyze its optimization dynamics and inference stability through the lenses of gradient rectification, manifold learning, and fixed-point theory.

**Privilege Learning as Gradient Rectification and Variance Reduction.** The core optimization challenge in all-in-one restoration stems from the "tug-of-war" dynamics where task-specific gradients conflict (i.e., $\cos(\nabla \mathcal{L}_i, \nabla \mathcal{L}_j) < 0$). We posit that the privileged information acts not merely as auxiliary supervision, but as a *Gradient Rectifier*. Let $g_{task}$ be the gradient from the degraded input and $g_{priv}$ be the gradient derived from the privileged loss. Since the privileged features represent a unified, degradation-free manifold, $g_{priv}$ provides a dominant descent direction positively correlated with the optimal solution for all tasks ($\langle g_{priv}, g_k^* \rangle > 0$). The addition of $g_{priv}$ constrains the optimization trajectory within a valid descent cone, effectively mitigating destructive interference. Furthermore, from a stochastic optimization perspective, $g_{priv}$ functions as a **Control Variate**. The randomness of degradations (e.g., rain streak locations) induces high variance in $g_{task}$. By fusing the deterministic privileged signal, we construct a gradient estimator with strictly lower variance, $\text{Var}(g_{SIPL}) < \text{Var}(g_{baseline})$, theoretically supporting the accelerated convergence observed in our experiments.

**Proxy Fusion as Sparse Manifold Projection.** We interpret the Proxy Fusion module geometrically as a projection operator onto the clean image manifold $\mathcal{M}$. The learned Privileged Dictionary, $PD = \{d_1, \ldots, d_N\}$, effectively spans the tangent space of $\mathcal{M}$. The cross-attention mechanism in Proxy Fusion performs a soft orthogonal projection:

$$\mathcal{P}_{\mathcal{M}}(F) = \text{Softmax}\left(\frac{F \cdot PD^T}{\sqrt{C}}\right) PD \tag{8}$$

This operation acts as a sparse approximation solver. By reconstructing the noisy feature $F$ using a linear combination of "clean" atoms from $PD$, the operator $\mathcal{P}_{\mathcal{M}}$ filters out degradation components (noise, artifacts) that lie in the orthogonal complement $\mathcal{M}^{\perp}$. This ensures that the refined features are explicitly constrained to the valid signal subspace, preventing error propagation.

**Stability of Iterative Self-Refinement.** A critical concern for feedback mechanisms is stability. We model the iterative inference as a discrete dynamical system $x_{t+1} = T(x_t)$, where $T$ is the refinement operator. Convergence to a stable fixed point $x^*$ is guaranteed if $T$ is a contraction mapping. In SIPL, the **frozen** nature of the Privileged Dictionary during inference is the key stabilizer. Since the dictionary atoms are fixed and bounded, the projection operator $\mathcal{P}_{\mathcal{M}}$ becomes non-expansive (Lipschitz constant $L \leq 1$). This effectively clips the spectral radius of the recurrent cycle, ensuring that the iterative process converges rapidly to a stable plateau (as evidenced in Table 6) rather than diverging into chaos. This distinguishes SIPL from naive self-feedback approaches, which often suffer from error amplification due to covariate shifts.

## B ADDITIONAL EXPERIMENTAL RESULTS AND ANALYSES

This section presents supplementary experimental results, including single-task performance benchmarks, out-of-distribution (OOD) generalization analyses, computational complexity comparisons, further iterative inference studies, and additional qualitative results.

### B.1 VISUALIZATION AND ANALYSIS OF THE PRIVILEGED DICTIONARY.

To demystify the role of the Privileged Dictionary (PD) and understand how it guides the restoration process, we conduct a comprehensive visualization analysis from two complementary perspectives: the intrinsic properties of the learned atoms and their dynamic influence on feature representations.

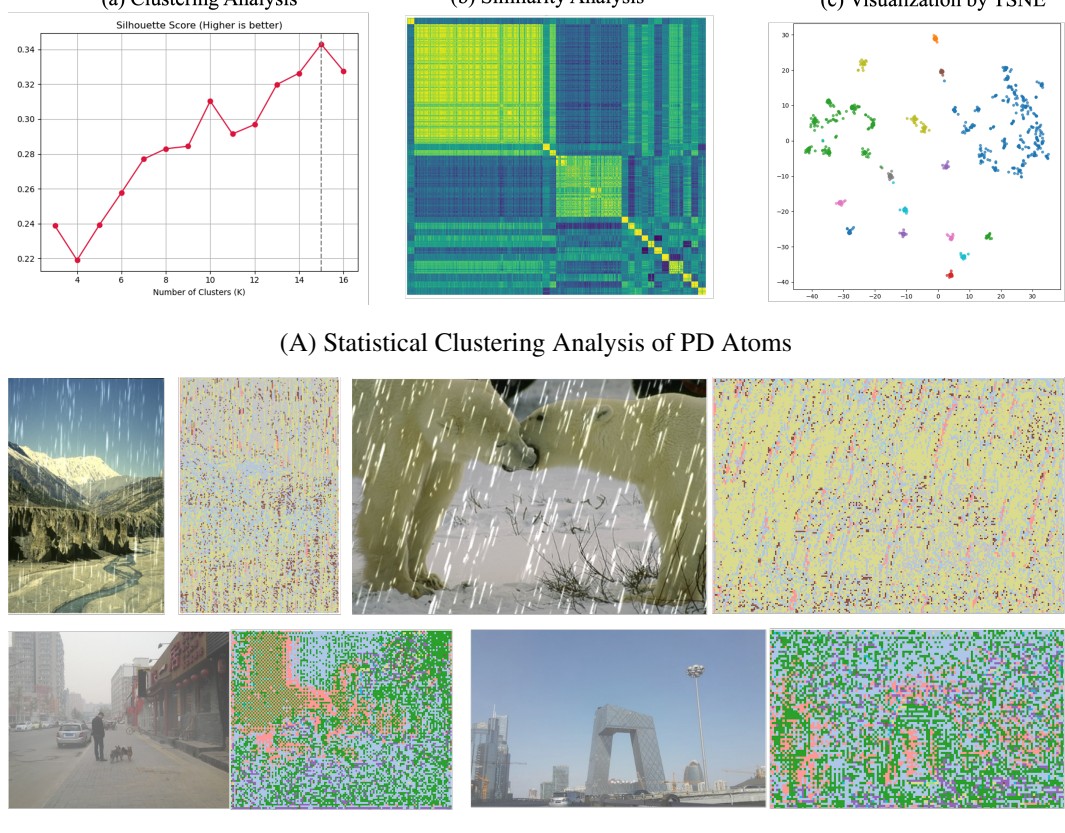

(A) Statistical Clustering Analysis of PD Atoms

(B) Visualization of Atom Usage (Physical Meaning)

Figure 10: **Analysis of the Privileged Dictionary (PD).** (A) Clustering metrics and t-SNE visualization reveal that the learned atoms form distinct, compact groups rather than a uniform distribution. (B) Mapping specific atoms back to the image space shows they correspond to physically meaningful structures, such as specific textures or edges.

**1) Intrinsic Properties and Physical Meaning.** Since the PD resides in a high-dimensional latent space, we first employ clustering analysis to interpret its structure. As shown in Figure 10(a), the Silhouette Score peaks around $K = 15$, suggesting that the learned atoms naturally group into distinct categories. The Similarity Matrix (Fig. 10(b)) and the t-SNE visualization (Fig. 10(c)) further corroborate this, revealing a compact and well-separated cluster structure. This indicates that the PD does not simply memorize random noise but captures specific prototype features. To uncover the physical meaning of these prototypes, we visualize the spatial distribution of the top-activated atoms during inference (Figure 10 bottom). We observe a strong semantic correlation: specific atoms consistently activate in regions with distinct structural characteristics, such as high-frequency textures, flat color blocks, or sharp edges. This confirms that the PD effectively functions as a codebook of degradation-agnostic structural priors, selecting appropriate basis functions to reconstruct diverse image contents.

**2) Impact on Feature Representation.** To verify the effectiveness of Proxy Fusion, we visualize the intermediate feature maps of the baseline PromptIR, our method at the initial pass (Iter-0), and the first refinement step (Iter-1) in Figure 11. Compared to the baseline, which exhibits diffuse attention often distracted by global context, our Iter-0 features demonstrate a significantly enhanced response to degradation patterns (e.g., clearly highlighting rain streaks and complex textures). This suggests that the PD, trained via Privilege Learning, successfully rectifies the feature extraction process to be more sensitive to high-frequency details. Furthermore, in the Iter-1 stage, where the Iter-0 output serves

as pseudo-privileged information, the feature focus shifts noticeably. As the background is largely restored in the initial pass, the attention mechanism in Iter-1 becomes more selective, specifically targeting residual artifacts and broken regions. This visual evolution empirically validates our core hypothesis: the PD acts as a stable anchor, enabling the model to iteratively refine the restoration from a coarse structural recovery to fine-grained detail correction.

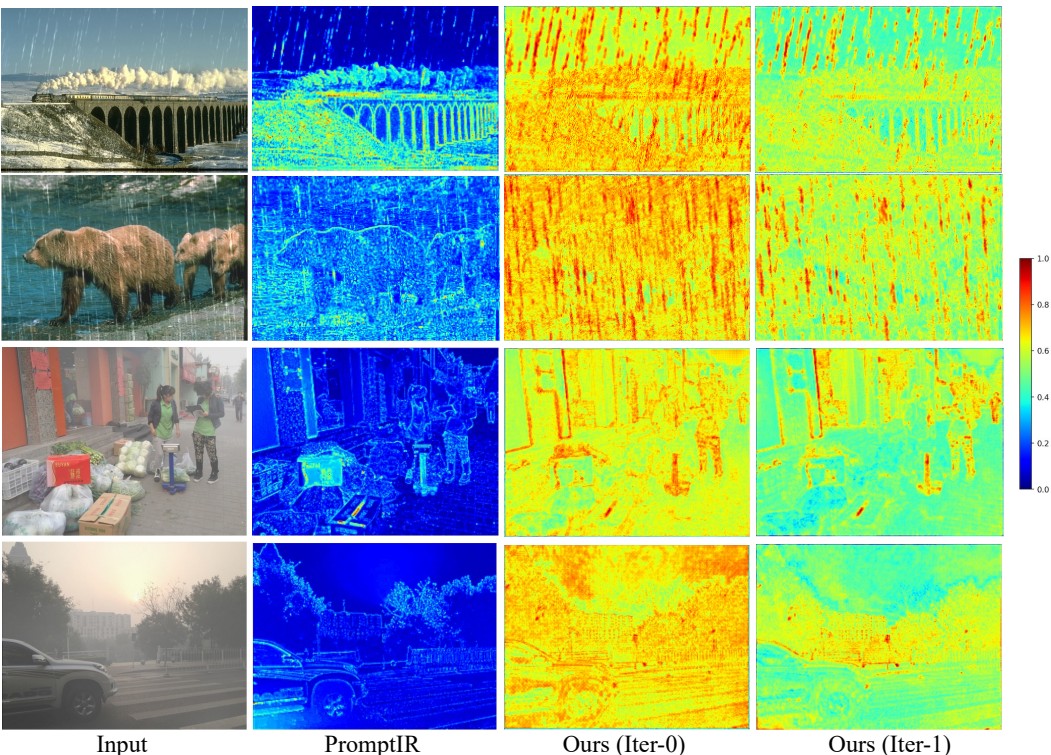

| Input | PromptIR | Ours (Iter-0) | Ours (Iter-1) |

Figure 11: **Evolution of Feature Representations.** Visual comparison of feature activations between the Baseline (PromptIR), SIPL (Iter-0), and SIPL (Iter-1). While the baseline's attention is diffuse, SIPL (Iter-0) effectively highlights rain streaks and textures. The subsequent refinement (Iter-1) further focuses on residual artifacts, demonstrating the effectiveness of our self-improvement mechanism.

### B.2 DETAILED ABLATION ON PRIVILEGED DICTIONARY SIZE

To determine the optimal capacity for the Proxy Fusion module, we conducted a fine-grained ablation study on the size of the Privileged Dictionary (PD), denoted as $N$. The dictionary size is a critical hyperparameter that governs the diversity and richness of the structural priors distilled from the ground truth. We evaluated the model's performance on the five-task benchmark by varying $N$ from 32 to 1024.

The detailed quantitative results are summarized in Table 8. We observe a clear positive correlation between the dictionary size and the overall restoration quality. Specifically, increasing $N$ from 32 to 256 yields steady improvements in average PSNR, rising from 30.34 dB to 30.53 dB. This trend validates our hypothesis that a larger dictionary capacity enables the model to retain a more comprehensive set of degradation-agnostic priors, such as high-frequency textures and sharp edges, which are essential for guiding the feature refinement process.

Notably, a significant performance leap is observed when scaling $N$ from 256 to 512, where the average PSNR boosts by **+0.26 dB** (from 30.53 dB to 30.79 dB). This suggests that a critical threshold of capacity is required to adequately cover the complex manifold of high-quality natural images across diverse scenes. However, further increasing $N$ to 1024 results in diminishing returns, with a marginal gain of only 0.03 dB in PSNR while doubling the dictionary parameters and increasing the

computational overhead of the attention mechanism. Consequently, considering the trade-off between performance maximization and computational efficiency, we adopt $N = 256$ as the default setting for our final model, as it achieves near-optimal performance with a reasonable resource footprint.

Table 8: Detailed ablation study on the impact of Privileged Dictionary (PD) size $N$. The results are reported on the Five-Task benchmark.

| Config | Denoising Noisy25 | Dehazing SOTS | Deraining Rain100L | Deblurring GoPro | Low-Light LOL | Average |
|---|---|---|---|---|---|---|
| 32 | 31.28 / 0.8882 | 30.12 / 0.9748 | 37.95 / 0.9801 | 29.01 / 0.8572 | 23.35 / 0.8549 | 30.34 / 0.9110 |
| 64 | 31.35 / 0.8895 | 30.25 / 0.9757 | 38.13 / 0.9813 | 29.12 / 0.8587 | 23.47 / 0.8558 | 30.46 / 0.9122 |
| 128 | 31.38 / 0.8902 | 30.09 / 0.9761 | 38.31 / 0.9825 | 29.36 / 0.8648 | 23.24 / 0.8560 | 30.48 / 0.9139 |
| 256 (Default) | 31.45 / 0.8880 | 30.51 / 0.9750 | 38.09 / 0.9820 | 29.35 / 0.8860 | 23.23 / 0.8560 | 30.53 / 0.9170 |
| 384 | 31.32 / 0.8887 | 30.68 / 0.9744 | 38.62 / 0.9860 | 29.02 / 0.8562 | 23.48 / 0.8561 | 30.62 / 0.9123 |
| 512 | 31.48 / 0.8923 | 30.75 / 0.9764 | 38.68 / 0.9830 | 29.42 / 0.8825 | 23.63 / 0.8577 | 30.79 / 0.9184 |
| 1024 | 31.50 / 0.8928 | 30.71 / 0.9768 | 38.62 / 0.9834 | 29.40 / 0.8831 | 23.68 / 0.8582 | 30.82 / 0.9189 |

### B.3 DETAILED EXPERIMENTAL SETUP

In this section, we provide the detailed experimental setup for the benchmarks used to evaluate our SIPL framework. We evaluate SIPL across four comprehensive all-in-one settings that encompass a wide range of real-world image degradation scenarios:

1. **Three-Task Setting:** Following the established protocol in Li et al. (2022); Potlapalli et al. (2023), we address three distinct degradation tasks: image denoising (with synthetic Gaussian noise), deraining (using the Rain100L dataset), and dehazing (on the RESIDE dataset). For denoising, we use BSD400 Arbeláez et al. (2011) and WED Ma et al. (2017) for training, and test on BSD68 with noise levels of 15, 25, and 50. For deraining, we employ the Rain100L Yang et al. (2017) dataset, and for dehazing, we use the outdoor subset of RESIDE Li et al. (2019a).

2. **Five-Task Setting:** To evaluate our model's capacity to handle a broader spectrum of degradations, we utilize a five-task benchmark, including deraining (Rain100L Yang et al. (2017)), dehazing (RESIDE Indoor Training Set), denoising (BSD400 + WED), motion deblurring (GoPro Nah et al. (2017)), and low-light enhancement (LOL Wei et al. (2018)). This setting follows Zhang et al. (2023) and tests the model's ability to handle diverse degradation types in real-world scenarios.

3. **Deweathering Setting:** Based on previous work Valanarasu et al. (2022), we use the AllWeather Valanarasu et al. (2022) dataset for training, containing images from Raindrop Qian et al. (2018), Outdoor-Rain Li et al. (2019b), and Snow100K Liu et al. (2018).

4. **Composite Degradation Setting:** Following the protocol established in Guo et al. (2024b), we evaluate our method on the Composite Degradation Dataset (CDD-11), which represents a more challenging scenario with mixed degradations. CDD-11 encompasses 11 categories of image degradations including single degradations (low-light, haze, rain, snow) and their combinations (low+haze, low+rain, low+snow, haze+rain, haze+snow, low+haze+rain, low+haze+snow). The dataset is constructed using standard benchmarks: the LOw-Light dataset (LOL) Wei et al. (2018), the REalistic Single Image DEhazing Outdoor Training Set (RESIDE-OTS) Li et al. (2019a), the Rain1200 dataset Zhang and Patel (2018), and the Snow100k dataset Liu et al. (2018). This setting particularly evaluates our framework's capability to handle complex, interacting degradations that better reflect real-world scenarios.

For all settings, we adopt the same training/testing splits and protocols as in the original works to ensure fair comparisons. We integrate our proposed SIPL framework into various backbone architectures to demonstrate its versatility and effectiveness.

## B.4 SINGLE-TASK PERFORMANCE EVALUATION

To further assess the efficacy of SIPL, we evaluated its performance on individual restoration tasks, aligning our experimental setup with that of PromptIR Potlapalli et al. (2023) and AdaIR Cui et al. (2025). These evaluations test the capability of our all-in-one model, enhanced with SIPL, on specialized degradation scenarios.

Table 9: Deraining results in the single-task setting on the Rain100L dataset. Our SIPL approach obtains a significant performance boost of 1.98 dB PSNR over baseline PromptIR and 0.12 dB over the AdaIR.

| Method | DIDMDN | UMR | SIRR | MSPFN | LPNet | AirNet | Restormer | PromptIR | AdaIR | **PromptIR + SIPL (Ours)** |
|--------|--------|-----|------|-------|-------|--------|-----------|----------|-------|---------------------------|
| PSNR | 23.79 | 32.39 | 32.37 | 33.50 | 33.61 | 34.90 | 36.74 | 37.04 | 38.90 | **39.02** |
| SSIM | 0.773 | 0.921 | 0.926 | 0.948 | 0.958 | 0.977 | 0.978 | 0.979 | 0.985 | **0.986** |

**Deraining on Rain100L:**   The Rain100L dataset Yang et al. (2017) serves as a standard benchmark for single-image deraining. As presented in Table 9, PromptIR augmented with our SIPL framework achieves state-of-the-art performance. Specifically, it obtains a PSNR of **39.02 dB** and an SSIM of **0.986**. This represents a substantial improvement of **1.98 dB** in PSNR over the original PromptIR baseline and also surpasses the strong AdaIR model by **0.12 dB**, demonstrating the significant benefits of SIPL in effectively removing rain streaks while preserving image fidelity.

**Dehazing on SOTS Outdoor:**   For evaluating dehazing performance, we utilize the outdoor test set from SOTS, part of the RESIDE dataset Li et al. (2019a). The results in Table 10 indicate that SIPL notably enhances PromptIR's dehazing capabilities. Our approach (PromptIR + SIPL) achieves a PSNR of **31.82 dB** and an SSIM of **0.982**. This is a gain of **0.51 dB** in PSNR compared to the PromptIR baseline. Furthermore, our method slightly outperforms AdaIR (31.80 dB PSNR / 0.981 SSIM), underscoring SIPL's efficacy in restoring clarity and detail in hazy conditions.

**Denoising using Five-Task Pre-trained Model:**   To assess robustness and generalization for denoising, we employed the all-in-one model pre-trained on five distinct degradation tasks (including denoising) and evaluated it directly on three commonly used denoising benchmark datasets: Urban100 Huang et al. (2015), Kodak24 Franzen (1999), and BSD68 Martin et al. (2001). This setup tests the model's ability to denoise effectively without task-specific fine-tuning. As detailed in Table 11, PromptIR + SIPL demonstrates superior performance, achieving an average PSNR of **31.98 dB** across all datasets and noise levels ($\sigma \in \{15, 25, 50\}$). This is a notable improvement over AdaIR (31.55 dB average PSNR). Particularly on the Urban100 dataset, which often contains complex structures, our method shows significant gains (e.g., **+1.29 dB** for $\sigma = 15$, **+1.06 dB** for $\sigma = 25$). Consistent, positive improvements are also observed across the Kodak24 and BSD68 datasets for all noise levels. These results, especially on datasets potentially unseen during the denoising phase of the five-task training, highlight the advanced robustness and generalization capabilities endowed by the SIPL framework.

**Enhancing All-in-One Model Transfer for Deblurring**   To evaluate how SIPL enhances a generalist model's capability on a specialist task, we fine-tuned our SIPL-enhanced PromptIR (originally trained on five tasks) for the single task of deblurring on the GoPro dataset. As shown in Table 12, our approach achieves a significant improvement over the baseline PromptIR and obtains a highly competitive performance against the recently proposed Perceive-IR. This result validates that the priors learned via SIPL improve the model's transferability, allowing it to effectively adapt and excel in dedicated, single-task scenarios.

Beyond validating SIPL's ability to enhance the transferability of all-in-one models, we further sought to evaluate its effectiveness in boosting a dedicated, high-performance single-task architecture. To this end, we integrated our SIPL framework into the strong NAFNet baseline and retrained it from scratch on two standard benchmarks: the GoPro dataset for deblurring and the SIDD dataset for denoising.

**Evaluation on Image Deblurring Task**   To further demonstrate SIPL's architectural agnosticism and effectiveness, we integrated it into a strong, dedicated single-task baseline, NAFNet, and retrained

Table 10: Quantitative comparison for single-task dehazing. Our SIPL achieves significant improvement over baseline PromptIR with 0.51 dB PSNR.

| Method | DehazeNet | MSCNN | AODNet | EPDN | FDGAN | AirNet | Restormer | PromptIR | AdaIR | **PromptIR + SIPL** |
|---|---|---|---|---|---|---|---|---|---|---|
| PSNR | 22.46 | 22.06 | 20.29 | 22.57 | 23.15 | 23.18 | 30.87 | 31.31 | 31.80 | **31.82** |
| SSIM | 0.851 | 0.908 | 0.877 | 0.863 | 0.921 | 0.900 | 0.969 | 0.973 | 0.981 | **0.982** |

Table 11: Image denoising results of directly applying the pre-trained model under the five-degradation setting to the Urban100 Huang et al. (2015), Kodak24 Franzen (1999) and BSD68 Martin et al. (2001) datasets. The results are PSNR scores. Our SIPL achieves significant improvement across all test datasets compared to previous SOTA method AdaIRCui et al. (2025).

| Method | Urban100 | | | Kodak24 | | | BSD68 | | | |
|---|---|---|---|---|---|---|---|---|---|---|
| | $\sigma = 15$ | $\sigma = 25$ | $\sigma = 50$ | $\sigma = 15$ | $\sigma = 25$ | $\sigma = 50$ | $\sigma = 15$ | $\sigma = 25$ | $\sigma = 50$ | Average |
| DL Fan et al. (2019) | 21.10 | 21.28 | 20.42 | 22.63 | 22.66 | 21.95 | 23.16 | 23.09 | 22.09 | 22.04 |
| Transweather Valanarasu et al. (2022) | 29.64 | 27.97 | 26.08 | 31.67 | 29.64 | 26.74 | 31.16 | 29.00 | 26.08 | 28.66 |
| TAPE Liu et al. (2022) | 32.19 | 29.65 | 25.87 | 33.24 | 30.70 | 27.19 | 32.86 | 30.18 | 26.63 | 29.83 |
| AirNet Li et al. (2022) | 33.16 | 30.83 | 27.45 | 34.14 | 31.74 | 28.59 | 33.49 | 30.91 | 27.66 | 30.89 |
| IDR Zhang et al. (2023) | 33.82 | 31.29 | 28.07 | 34.78 | 32.42 | 29.13 | 34.11 | 31.60 | 28.14 | 31.48 |
| AdaIR Cui et al. (2025) | 34.10 | 31.68 | 28.29 | 34.89 | 32.38 | 29.21 | 34.01 | 31.35 | 28.06 | 31.55 |
| **PromptIR + SIPL** | **35.39** | **32.74** | **29.17** | **34.98** | **32.50** | **29.36** | **34.08** | **31.45** | **28.16** | **31.98** |

it from scratch on the GoPro dataset. The results in Table 13 show that our PL-enhanced training (Iter-0) already provides a better starting model than the original NAFNet. The subsequent self-refinement step (Iter-1) yields an additional performance boost, setting a new state-of-the-art result. This confirms that SIPL is not limited to enhancing all-in-one models but also serves as a general and effective framework for pushing the performance of specialized, high-performing architectures.

**Evaluation on Image Denoising Task**  To validate SIPL's versatility across different tasks, we applied it to the single-task denoising benchmark on the SIDD dataset, again using NAFNet as the backbone. As presented in Table 14, the results are consistent with our other findings. The PL-enhanced training (Iter-0) establishes a stronger baseline model compared to the original NAFNet. Subsequently, the inference-time self-refinement step (Iter-1) provides a further boost, achieving a final PSNR of 40.19 dB. These comprehensive single-task results strongly validate that SIPL is an efficient and general framework for improving image restoration models across diverse tasks, datasets, and backbone architectures.

**Evaluation on Real-World Deraining**   We conducted experiments on the **SPA** (Spatial Attention) dataset, a widely recognized large-scale benchmark for real-world image deraining. Real-world rain streaks often exhibit complex variations in density, direction, and illumination that are difficult to simulate perfectly with synthetic data. To demonstrate the effectiveness of our method, we compared our SIPL-enhanced models against a comprehensive set of state-of-the-art methods, including Restormer, IDT, and Uformer. We integrated SIPL into two strong baselines: DRSformer and AST.

The quantitative comparisons are summarized in Table 15. As observed, recent transformer-based methods like Uformer and DRSformer have already established high performance baselines. However, by incorporating the SIPL framework, we achieve further tangible improvements. Specifically, SIPL boosts the performance of DRSformer to **49.07 dB** and AST to **49.92 dB**, surpassing competing approaches. These gains suggest that the Privileged Dictionary effectively captures degradation-agnostic structural priors during training, which then guide the model to better distinguish between complex rain artifacts and genuine image details during inference, thereby enhancing robustness against the domain gap inherent in real-world data.

**Evaluation on Real-World Low-Light Enhancement**   We further evaluated our method on the **LOL-v2 (Real)** dataset for low-light image enhancement. This dataset consists of real-world low-light/normal-light pairs characterized by severe noise distributions and complex color shifts, presenting a significant challenge for restoration models compared to synthetic data. We compared our approach against established low-light enhancement methods such as KinD, MIRNet, and SNR-Net, using Restormer and Retinexformer as our primary backbones.

Table 12: Performance of all-in-one models on the single task of deblurring (GoPro dataset). PSNR/SSIM values are reported. Our SIPL-enhanced model demonstrates superior transferability.

| Method
Venue | AirNet Li et al. (2022)
CVPR'22 | PromptIR Potlapalli et al. (2023)
NeurIPS'23 | Perceive-IR
IEEE TIP'25 | **PromptIR + SIPL (Ours)**
2025 |
|---|---|---|---|---|
| **PSNR / SSIM** | 31.64 / 0.945 | 32.41 / 0.956 | **32.83 / 0.960** | 32.77 / 0.961 |

Table 13: Single-task state-of-the-art comparison for deblurring on the GoPro dataset. SIPL provides a clear performance boost to the strong NAFNet baseline.

| Method
Venue | Restormer
CVPR'22 | UFormer
CVPR'22 | MaIR
ECCV'22 | NAFNet
ECCV'22 | NAFNet+SIPL(Iter-0)
2025 | **NAFNet+SIPL(Iter-1)**
**2025** |
|---|---|---|---|---|---|---|
| **PSNR / SSIM** | 32.92 / 0.961 | 33.06 / 0.967 | 33.69 / 0.969 | 33.69 / 0.966 | 33.76 / 0.968 | **33.82 / 0.970** |

Table 16 presents the comparative results on the real-world subset of LOL-v2. While specialized methods like SNR-Net and Retinexformer perform strongly, integrating SIPL yields consistent performance gains. Notably, the SIPL-enhanced Retinexformer achieves the highest PSNR of **23.21 dB** and SSIM of **0.847**. These results indicate that our iterative self-refinement mechanism is particularly effective in handling the complex noise and illumination artifacts found in real-world low-light scenarios, leading to reconstructions with higher fidelity and better visual quality.

**Summary of Single-Task Evaluations.** While SIPL is designed to excel in challenging all-in-one settings, these comprehensive single-task evaluations confirm its broad effectiveness and versatility. The experiments demonstrate two key strengths: first, SIPL significantly enhances the transferability of a generalist model (PromptIR) to specialized tasks. Second, it provides a substantial performance boost to a dedicated, high-performance single-task model (NAFNet) trained from scratch. The consistent improvements across diverse tasks like deraining, dehazing, and denoising validate that SIPL is a truly task- and model-agnostic framework, effectively improving restoration capabilities in both multi-task and single-task scenarios.

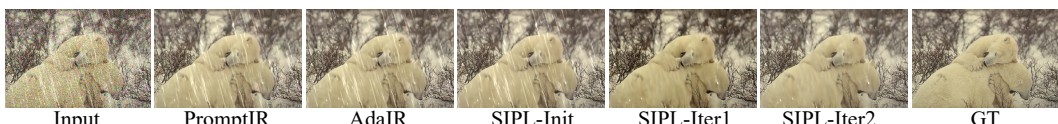

| Input | PromptIR | AdaIR | SIPL-Init | SIPL-Iter1 | SIPL-Iter2 | GT |

Figure 12: Visual illustration of OOD performance on a challenging Rain100L + Gaussian Noise ($\sigma = 50$) example. From left to right: Degraded Input, PromptIR, AdaIR, SIPL (Initial), SIPL (Iter-1), SIPL (Iter-2), and Ground Truth (GT). The iterative application of SIPL progressively enhances image clarity, restores fine details, and reduces artifacts, significantly outperforming baseline methods and demonstrating effective generalization to unseen composite degradations.

## B.5 Out-of-Distribution (OOD) Generalization Analysis

A critical attribute of advanced image restoration models is their capacity to generalize effectively to previously unseen degradation types. This section investigates the OOD generalization capabilities of our SIPL-enhanced framework, drawing comparisons with established methods like PromptIR and AdaIR. Our analysis specifically focuses on performance when encountering complex, composite degradations not present during the training phase.

**Performance on Unseen Composite Degradation (Rain100L + Noise):** We evaluate models originally trained on three distinct restoration tasks (deraining, dehazing, denoising individually) on a challenging synthetic dataset: Rain100L combined with varying levels of Gaussian noise ($\sigma \in \{15, 25, 50\}$). This composite degradation scenario was deliberately excluded from the training set to rigorously test OOD performance.

The quantitative results are presented in Table 17. Baseline models, PromptIR and AdaIR, achieve average PSNR scores of 24.40 dB and 24.39 dB, respectively, on this unfamiliar task. Our PromptIR

Table 14: Single-task denoising results on the SIDD validation set. Our SIPL framework again demonstrates its value by improving the performance of the NAFNet baseline.

| Method | MAXIM | CGNet | NAFNet | NAFNet+SIPL(Iter-0) | NAFNet+SIPL(Iter-1) |
| Venue | CVPR'22 | TMLR'24 | ECCV'22 | 2025 | **2025** |
|---|---|---|---|---|---|
| **PSNR (dB)** | 40.02 | 40.39 | 39.93 | 40.10 | **40.19** |

Table 15: Quantitative comparison on the **SPA** real-world deraining dataset. The proposed SIPL framework further elevates the performance of advanced baselines (DRSformer and AST) beyond existing methods.

| Method | Restormer | SCD-Former | IDT | Uformer | DRSformer | **+SIPL** | AST | **+SIPL** |
|---|---|---|---|---|---|---|---|---|
| **PSNR ↑** | 46.25 | 46.89 | 47.34 | 47.84 | 48.53 | **49.07** | 49.51 | **49.92** |
| **SSIM ↑** | 0.9911 | 0.9941 | 0.9929 | 0.9925 | 0.9924 | **0.9941** | 0.9942 | **0.9953** |

+ SIPL model, in its initial single-pass inference ("PromptIR + SIPL (initial)"), yields a comparable average PSNR of 24.46 dB. However, the transformative advantage of SIPL becomes strikingly evident through its iterative self-improvement mechanism. With just one iteration ("+Iter 1"), the average PSNR significantly jumps to 25.48 dB. A second iteration ("+Iter 2") further elevates the performance dramatically to an average PSNR of **29.23 dB**. This represents a remarkable **+4.77 dB** improvement over its initial state and far surpasses the static performance of the baseline models. This progressive and substantial enhancement underscores the robust OOD generalization conferred by SIPL, particularly its ability to iteratively refine results when faced with novel degradations.

The qualitative improvements are visualized in Figure 12 using an example from the Rain100L + Noise ($\sigma = 50$) set. While the input image exhibits significant degradation, and baseline methods like PromptIR and AdaIR offer limited restoration, our SIPL demonstrates clear visual enhancements. The initial output ("SIPL-Init") shows some improvement, but subsequent iterations ("SIPL-Iter1", "SIPL-Iter2") progressively recover finer details, enhance sharpness, and reduce artifacts more effectively, approaching the ground truth quality. This visual evidence corroborates the quantitative gains and highlights the practical efficacy of iterative refinement in complex OOD scenarios.

**Efficacy of Self-Improvement in OOD Contexts:** The marked success of PromptIR + SIPL in handling these unseen composite degradations, especially through iteration, is attributable to its core design featuring the Privileged Dictionary (PD) and the self-improvement learning strategy. Unlike baseline models such as PromptIR and AdaIR, which are not inherently designed to leverage their own outputs for iterative refinement on OOD tasks without a guiding mechanism, SIPL excels in this regard. Standard architectures, if naively iterated on OOD inputs, might see performance stagnate or even degrade due to the accumulation of errors or model biases when processing unfamiliar data distributions.

In stark contrast, SIPL's PD, trained to distill essential characteristics of high-quality images, provides robust guidance even when the pseudo-privileged information is derived from an imperfectly restored OOD image. The iterative process allows the model to progressively correct errors and enhance image quality by repeatedly consulting these learned priors. This capacity for effective self-correction and refinement in the face of novel, complex degradations is a key differentiator of our approach.

This OOD analysis strongly suggests that our self-improved iteration paradigm offers a novel and potent pathway for advancing all-in-one image restoration. Beyond striving for optimal performance in a single forward pass, SIPL demonstrates the significant potential of empowering models to adapt and improve their outputs dynamically at test time. This is particularly crucial for real-world scenarios where diverse and unforeseen degradations are common, showcasing a promising direction for developing more versatile and robust restoration solutions.

Table 16: Quantitative comparison on the **LOL-v2 (Real)** dataset. Our SIPL-enhanced models demonstrate superior restoration quality compared to existing state-of-the-art methods in challenging real-world lighting conditions.

| Method | KinD | MIRNet | SNR-Net | FourLLIE | Restormer | +SIPL | Retinexformer | +SIPL |
|---|---|---|---|---|---|---|---|---|
| **PSNR ↑** | 14.74 | 20.02 | 21.48 | 21.60 | 19.94 | **20.44** | 22.80 | **23.21** |
| **SSIM ↑** | 0.641 | 0.820 | 0.849 | 0.847 | 0.827 | **0.838** | 0.840 | **0.847** |

Table 17: OOD performance on Rain100L with Gaussian noise ($\sigma =$15, 25, and 50). Models were trained on three distinct tasks and test on this unseen dataset directly. Iter-N denotes N iterative refinement steps. Best results are underlined, our method is highlighted.

| Method | Rain100L + Gaussian Noise | | | Average |
|---|---|---|---|---|
| | $\sigma = 15$ | $\sigma = 25$ | $\sigma = 50$ | |
| PromptIR Potlapalli et al. (2023) | 24.92 | 24.50 | 23.79 | 24.40 |
| AdaIR Cui et al. (2025) | 24.91 | 24.50 | 23.77 | 24.39 |
| **PromptIR + SIPL (initial)** | **24.95** | **24.59** | **23.86** | **24.46** |
| **+ Iter-1** | **26.90** | **25.59** | **23.96** | **25.48** |
| **+ Iter-2** | **31.74** | **29.53** | **26.41** | **29.23** |

## B.6 GENERALIZATION ANALYSIS AND INFERENCE STABILITY

Beyond standard benchmarks, a robust restoration framework must demonstrate the capability to generalize to unseen degradation types and maintain stability under adverse conditions. We investigate these attributes from two complementary perspectives: quantitative cross-dataset validation and qualitative failure case analysis.

**Generalization to Unseen Composite Degradations.** To rigorously assess the model's performance on degradation patterns absent from the training data, we conducted an Out-Of-Distribution (OOD) evaluation using the **CDD-11 dataset**. The model, trained solely on the three-task setting (Deraining, Dehazing, Denoising), was directly tested on complex composite degradations (e.g., Low-light+Haze, Haze+Rain) without any fine-tuning. As presented in Table 18, our SIPL framework consistently outperforms the PromptIR baseline across all 11 unseen degradation categories. For instance, in the challenging "Haze+Rain" scenario, SIPL achieves a PSNR gain of **+0.52 dB**. This quantitative evidence suggests that the Privileged Dictionary captures universal, high-quality structural priors that transcend specific degradation types, enabling the model to effectively regularize features even when the input distribution shifts significantly.

Table 18: Evaluation on CDD-11 Datasets using the 5-task model. SIPL demonstrates superior robustness on unseen composite degradations compared to the baseline PromptIR.

| Method | h | r | s | l+h | l+r | l+s | h+r | h+s | l+h+r | l+h+s | Average |
|---|---|---|---|---|---|---|---|---|---|---|---|
| PromptIR | 12.14 | 21.51 | 23.70 | 14.94 | 14.89 | 13.68 | 19.08 | 18.56 | 15.33 | 14.67 | 15.31 |
| **Ours** | **12.95** | **22.41** | **24.30** | **15.47** | **15.69** | **14.31** | **19.60** | **19.13** | **15.87** | **14.92** | **15.87** |

**Stability Analysis in Failure Cases.** While SIPL improves generalization, it is equally important to analyze its behavior when faced with intractable degradations. We visualize a failure case in Figure 13 involving *heavy snow*, a degradation type characterized by large-scale white occlusions that severely destroy image content and was not seen during training. As observed, neither the baseline nor our method can perfectly remove the heavy snow due to the massive information loss. However,

a critical observation lies in the **stability** of the iterative process. Unlike naive pixel-level feedback loops which often hallucinate artifacts or cause color collapse when processing OOD inputs, our feature-level self-refinement remains robust. The iterative updates (Iter-1) do not amplify errors or introduce new artifacts in the occluded regions. This confirms that the Proxy Fusion module acts as a "conservative" projection operator: if the degraded features cannot find a strong match in the clean Privileged Dictionary, the attention mechanism avoids forcing an erroneous reconstruction, thereby preventing performance collapse and ensuring a safe inference trajectory.

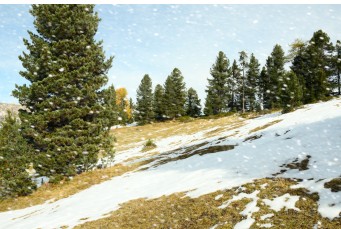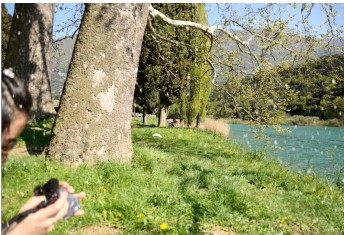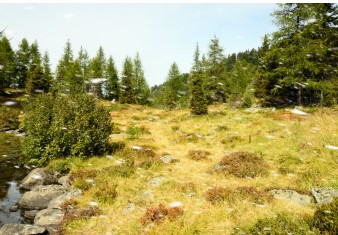

Figure 13: **Stability Analysis on Unseen Heavy Snow.** Visual comparison on the unseen heavy snow image. While the massive occlusion limits the restoration quality for all methods, SIPL demonstrates remarkable stability. The iterative refinement does not lead to artifact amplification or mode collapse, validating the robustness of our feature-space projection mechanism.

### B.7 ANALYSIS OF EFFICIENCY AND PERFORMANCE TRADE-OFFS

A core aspect of the SIPL framework is its flexibility, offering different trade-offs between computational cost and restoration quality. In this section, we provide a comprehensive analysis of this trade-off, first by dissecting the costs of SIPL's components on a large-scale model (PromptIR), and second, by comparing SIPL's iterative refinement against the common strategy of brute-force model scaling (on NAFNet).

**Cost-Benefit Analysis on PromptIR.** As summarized in Table 19, our methodology provides a spectrum of enhancements. The foundational Privilege Learning (PL), as a training-only strategy, offers a zero-cost inference boost, improving the PromptIR baseline by +0.9 dB on the five-task benchmark without any additional parameters or FLOPs. Building on this, our full SIPL framework in a single-pass configuration (Iter-0) adds a marginal cost (3M parameters and 20G FLOPs) for a further performance increase to 30.17 dB. This demonstrates the high efficiency of the Proxy Fusion module in retaining and applying privileged knowledge.

The full potential of SIPL is unlocked via iterative refinement (Iter-1), which, while increasing the computational load to 434G FLOPs, delivers the highest performance at 30.53 dB. We acknowledge this increased cost. However, this configuration represents a valuable trade-off, providing a powerful mechanism for tackling the most challenging restoration scenarios (e.g., composite and OOD degradations) where single-pass models often fall short.

Table 19: Comparison of model parameters and computational complexity for the PromptIR backbone on the five-task benchmark. FLOPs are calculated for a $256 \times 256$ input.

| Method | Parameters | FLOPs | Avg. PSNR (dB) |
|---|---|---|---|
| AirNet Li et al. (2022) | 9M | 301G | 25.44 |
| Transweather Valanarasu et al. (2022) | 21.5M | 115.2G | 25.22 |
| PromptIR Potlapalli et al. (2023) | 36M | 173G | 29.15 |
| AdaIR Cui et al. (2025) | 29M | 162G | 30.20 |
| **PromptIR + PL** | 36M | 173G | 30.05 |
| **PromptIR + SIPL** (Iter-0) | 39M | 193G | 30.17 |
| **PromptIR + SIPL** (Iter-1) | 39M | 434G | 30.53 |

**SIPL as an Efficient Alternative to Model Scaling.** To further contextualize SIPL's efficiency, we investigate a crucial question: is it better to apply SIPL to a smaller model or to simply train a

larger one (such as NAFNet-32+SIPL vs NAFNet-64)? To answer this, we compare the performance and cost of applying SIPL to NAFNet-32 against a much larger NAFNet-64 baseline on the SIDD denoising task.

Table 20: Performance vs. FLOPs comparison on the SIDD benchmark. SIPL on a smaller model (NAFNet-32) nearly matches the performance of a much larger model (NAFNet-64) with significantly less computation.

| Method | PSNR (dB) on SIDD | FLOPs |
|---|---|---|
| NAFNet-32 (baseline) | 39.96 | 16.08 G |
| NAFNet-64 (larger model) | 40.30 | 63.28 G |
| **NAFNet-32 + SIPL** (Iter-0) | 40.10 | 17.63 G |
| **NAFNet-32 + SIPL** (Iter-1) | **40.19** | **43.93 G** |

The results in Table 20 are compelling. NAFNet-32 + SIPL (Iter-1) achieves a PSNR of 40.19 dB, closing nearly all of the performance gap to the much larger NAFNet-64 (40.30 dB). Critically, it does so while requiring approximately **31% fewer FLOPs** (43.93G vs. 63.28G). This analysis compellingly demonstrates that SIPL is not merely an add-on for performance gain; it represents a more computationally efficient strategy for achieving top-tier results than simply scaling up a model's architecture. It validates that intelligently distilling and reusing privileged priors provides a more robust and efficient path to improvement.

**Concluding Remarks.** In summary, our efficiency analysis positions SIPL as a highly practical and versatile framework. It offers practitioners a flexible toolkit: from a zero-cost training enhancement (PL) to a powerful iterative refinement that can serve as a more efficient alternative to training ever-larger models. This makes SIPL a valuable contribution for developing high-performance image restoration solutions within specific computational budgets.

## B.8 COMPARISON WITH TEST-TIME ADAPTATION (TTA)

To rigorously situate our Self-Improved Privilege Learning (SIPL) framework within the broader landscape of inference-time strategies, we provide a detailed comparison with the Test-Time Adaptation (TTA) paradigm. While both approaches operate during the testing phase, they are fundamentally distinct in their objectives, mechanisms, and underlying philosophies.

**Conceptual Distinctions.** Following the definition in a recent comprehensive survey **?**, TTA is characterized as a paradigm to "adapt a pre-trained model to unlabeled data during testing... by training self-adapted models prior to inference." This definition highlights two core tenets of TTA: (1) its primary goal is *adaptation* to mitigate distribution shifts (domain gaps), and (2) its mechanism typically involves *updating model parameters* via gradient-based optimization on the test data.

In contrast, SIPL is designed as an *enhancement* framework rather than an adaptation technique. Its primary objective is to improve the restoration quality of a generalist model by mitigating inter-task conflicts inherent in multi-degradation learning. Crucially, SIPL operates on a **frozen** model during inference. It does not perform parameter updates or learn from the test data stream; instead, it retrieves and reuses degradation-agnostic priors distilled into the Privileged Dictionary (PD) during the training phase. The fundamental differences are summarized in Table 21.

Table 21: Conceptual Comparison between Test-Time Adaptation (TTA) and SIPL.

| Feature | Typical Test-Time Adaptation (TTA) | Self-Improved Privilege Learning (SIPL) |
|---|---|---|
| **Primary Goal** | **Adaptation**: Mitigate domain shift (Sim-to-Real). | **Enhancement**: Improve restoration quality via priors. |
| **Mechanism** | **Gradient Update**: Finetuning parameters on test data. | **Gradient-Free**: Forward pass with frozen dictionary. |
| **Knowledge Source** | Derived *online* from the test stream itself. | Derived *offline* from Ground Truth (Privileged Information). |

**Empirical Comparison and Efficiency Analysis.** To quantify the practical differences, we implemented a standard TTA baseline using the PromptIR backbone. The TTA process involved generating pseudo-labels for test inputs and fine-tuning the model parameters using an $\ell_1$ loss for 3,000 iterations per image.

The comparative results on the three-task benchmark are presented in Table 22. Standard TTA yields a marginal improvement (+0.03 dB) over the baseline. This limited gain is expected, as the test data in this benchmark follows a similar distribution to the training set, leaving little room for domain adaptation to act. In contrast, SIPL achieves a significant performance boost of **+0.61 dB**. This indicates that the benefit of SIPL stems from the explicit injection of structural priors via the Privileged Dictionary, rather than distribution alignment.

Furthermore, regarding computational efficiency, SIPL requires only 2 forward passes (for a single refinement step), whereas the gradient-based TTA baseline requires thousands of forward and backward passes. Thus, SIPL offers a more practical trade-off for real-time applications where latency is a constraint.

Table 22: Performance and Efficiency Comparison: SIPL vs. Gradient-based TTA on the Three-Task Benchmark.

| Method | Mechanism | Inference Cost | Avg. PSNR (dB) |
|---|---|---|---|
| Baseline (PromptIR) | One-pass Inference | $1\times$ | 32.06 |
| TTA (Full Tuning) | Gradient Update | $\sim 3000\times$ | 32.09 (+0.03) |
| **SIPL (Ours)** | **Feature Refinement** | **$2\times$** | **32.67** (+0.61) |

**Complementarity of SIPL and TTA.** Finally, we investigated whether SIPL and TTA are mutually exclusive. By applying TTA techniques to fine-tune the Privileged Dictionary of a pre-trained SIPL model on test data, we observed further performance gains (e.g., increasing the average PSNR to 32.74 dB). This suggests that SIPL's structural priors and TTA's domain adaptation capabilities are complementary. While TTA aligns the model to the specific statistics of the test instance, SIPL ensures the restoration remains grounded in the manifold of high-quality natural images.

### B.9 QUANTITATIVE RESULTS OF MULTI-STEP REFINEMENT

This section provides a comprehensive analysis of the behavior of our Self-Improved Privilege Learning (SIPL) framework across multiple iterative refinement steps. While the main paper establishes that a single iteration ($t = 1$) offers an optimal trade-off between performance and efficiency, a deeper examination of the iterative process reveals crucial insights into the stability and robustness of our proposed mechanism. We present detailed quantitative results for both the Three-Task and Five-Task benchmarks over ten iterations, followed by an in-depth discussion.

Tables 23 and 24 detail the performance progression from the first to the tenth refinement step ($t = 10$). We also include the performance of a "GT-guided" oracle, where the ground-truth image is used as privileged information during inference, representing a practical upper bound for our method.

**Rapid Initial Improvement and Graceful Convergence** A consistent pattern emerges from both benchmarks: the vast majority of the performance improvement is achieved within the very first iteration ($t = 1$). On the Five-Task benchmark, this initial step accounts for a +1.38 dB PSNR gain, which is over 95% of the total improvement observed towards the peak performance at $t = 4$ (+1.45 dB). This indicates that our Proxy Fusion mechanism is highly effective at providing a strong initial correction signal.

Beyond the first step, the performance exhibits graceful convergence, quickly settling onto a stable plateau. The fluctuations in subsequent iterations are minimal (e.g., within $\pm 0.004$ dB on the

Table 23: Detailed performance metrics (PSNR) across ten iterations on the **Three-Task Benchmark**. The performance rapidly improves at $t = 1$ and then converges to a stable plateau, demonstrating the robustness of the self-refinement process. The average PSNR is shown with three decimal places to highlight the subtle changes in later iterations.

| Method | Average | Deraining | Dehazing | Denoising ($\sigma = 15$) | Denoising ($\sigma = 25$) | Denoising ($\sigma = 50$) |
|---|---|---|---|---|---|---|
| PromptIR (Baseline) | 32.060 | 36.37 | 30.58 | 33.98 | 31.31 | 28.06 |
| +SIPL (Iter-1) | 32.669 | 38.431 | 31.092 | 34.119 | 31.481 | 28.220 |
| +SIPL (Iter-2) | 32.670 | 38.428 | 31.095 | 34.122 | 31.482 | 28.225 |
| +SIPL (Iter-3) | 32.673 | 38.432 | 31.096 | 34.123 | 31.484 | 28.228 |
| +SIPL (Iter-4) | **32.673** | **38.433** | 31.098 | 34.122 | 31.483 | **28.230** |
| +SIPL (Iter-5) | **32.673** | 38.432 | **31.101** | 34.121 | 31.482 | 28.229 |
| +SIPL (Iter-6) | 32.671 | 38.431 | 31.097 | 34.120 | 31.481 | 28.227 |
| +SIPL (Iter-7) | 32.670 | 38.430 | 31.096 | 34.120 | 31.480 | 28.226 |
| +SIPL (Iter-8) | 32.669 | 38.429 | 31.094 | 34.119 | 31.480 | 28.225 |
| +SIPL (Iter-9) | 32.669 | 38.428 | 31.093 | 34.119 | 31.479 | 28.224 |
| +SIPL (Iter-10) | 32.669 | 38.428 | 31.094 | 34.118 | 31.480 | 28.224 |
| with GT (Upper Bound) | 33.099 | 38.792 | 31.551 | 34.615 | 31.838 | 28.701 |

Table 24: Detailed performance metrics (PSNR) across ten iterations on the **Five-Task Benchmark**. Similar to the three-task setting, the most significant gain is achieved in the first step, followed by stable convergence.

| Method | Average | Deraining | Dehazing | Denoising | Deblurring | Low-Light |
|---|---|---|---|---|---|---|
| PromptIR (Baseline) | 29.15 | 36.37 | 26.54 | 31.47 | 28.71 | 22.68 |
| +SIPL (Iter-1) | 30.53 | 38.09 | 30.51 | 31.45 | 29.35 | 23.23 |
| +SIPL (Iter-2) | 30.58 | 38.22 | 30.52 | 31.50 | **29.39** | 23.26 |
| +SIPL (Iter-3) | 30.58 | 38.24 | 30.52 | **31.53** | 29.37 | 23.26 |
| +SIPL (Iter-4) | **30.60** | **38.38** | **30.53** | 31.50 | 29.37 | 23.21 |
| +SIPL (Iter-5) | 30.58 | 38.28 | **30.53** | 31.48 | 29.34 | **23.28** |
| +SIPL (Iter-6) | 30.55 | 38.10 | 30.54 | 31.46 | 29.35 | 23.24 |
| +SIPL (Iter-7) | 30.53 | 38.08 | 30.53 | 31.48 | 29.34 | 23.24 |
| +SIPL (Iter-8) | 30.54 | 38.15 | 30.52 | 31.47 | 29.33 | 23.23 |
| +SIPL (Iter-9) | 30.52 | 38.07 | 30.51 | 31.44 | 29.35 | 23.22 |
| +SIPL (Iter-10) | 30.53 | 38.09 | 30.52 | 31.45 | 29.35 | 23.22 |
| with GT (Upper Bound) | 30.82 | 38.40 | 30.60 | 31.54 | 29.43 | 24.11 |

Three-Task average PSNR after $t = 2$). This is a critical finding. As demonstrated in our main paper's ablation study, a naive iterative loop (feeding the output pixels back as input) leads to catastrophic performance degradation due to the amplification of artifacts from out-of-distribution inputs. The stability of SIPL empirically proves that our feature-level refinement, guided by the Privileged Dictionary (PD), is a fundamentally different and robust process. The PD acts as a constant, degradation-agnostic anchor, ensuring the iterative process does not diverge but instead converges towards the learned manifold of high-quality images.

**Understanding Performance Saturation and the Role of Pseudo-PI** The performance saturation observed in later iterations is an expected and desirable behavior. Our iterative process relies on the model's own output as *pseudo-privileged information* (pseudo-PI). While this output is significantly cleaner than the original degraded input, it is still imperfect. The model iteratively refines its output until the pseudo-PI is no longer informative enough to elicit further significant improvements from the PD. The remaining performance gap between the saturated performance (e.g., 30.60 dB on Five-Task) and the GT-guided upper bound (30.82 dB) quantifies the inherent limitation of using a pseudo-guide instead of a perfect one. This gap highlights a potential avenue for future research in improving the fidelity of the pseudo-PI.

**Justification for Single-Step Refinement in Main Experiments**    This detailed analysis provides a strong empirical foundation for our choice of a single refinement step ($t = 1$) as the default configuration in the main paper. This single step captures the most substantial portion of the performance gain while being computationally efficient (requiring only two forward passes). While further iterations can yield marginal improvements, the diminishing returns suggest that a single, powerful refinement step offers the most practical and compelling trade-off between restoration quality and inference latency.

### B.10    ETHICS STATEMENT

This work is in full compliance with the ICLR Code of Ethics. Our research proposes a foundational algorithm for low-level image enhancement, aiming to contribute to the scientific community by providing a more effective and efficient solution.

We uphold the principles of scientific excellence and transparency. All datasets utilized in our experiments are publicly available and open-source, and we provide a detailed description of our methodology and experimental setup to ensure reproducibility. As a computer vision algorithm, our work does not directly involve personal data or raise immediate concerns regarding privacy or societal fairness. We have, however, transparently discussed the potential limitations of our method in Section 4.3 of this paper. While we have not identified direct negative societal impacts, we acknowledge that as a foundational technology, its subsequent applications should be developed and deployed responsibly by others. We declare no competing interests in this work.

### B.11    REPRODUCIBILITY STATEMENT

To ensure the reproducibility of our findings, we provide comprehensive documentation of our methodology, experimental setup, and resources. The complete source code, including scripts to reproduce all experiments and results presented in this paper, is available in the supplementary materials. A detailed description of our proposed model and algorithm is provided in Section 3 of the main paper. Further implementation details, including all hyperparameter settings, library dependencies, and the computational environment used for our experiments, are thoroughly documented in Appendix.

### B.12    THE USE OF LARGE LANGUAGE MODELS

The authors affirm that Large Language Models (LLMs) were not used for the core scientific contributions of this work. Specifically, LLMs were not utilized in the ideation phase, for the development of the proposed methodology, in the design or execution of experiments, or for the analysis of results. The conclusions presented in this paper were drawn entirely by the authors.

Following the completion of the main manuscript, an LLM-based tool was used as a general-purpose writing assistant. Its role was strictly limited to performing grammar checks and providing suggestions for improving the clarity and flow of sentences to enhance readability. The scientific content and integrity of the paper were not influenced by the use of the LLM.

