# OpenReview forum: "Self-Improved Prior for All-in-One Image Restoration"
_ICLR.cc/2026/Conference — Submitted to ICLR 2026_

### Official Review · Reviewer_n3ui · 2025-10-28

**Soundness:** 3
**Presentation:** 3
**Contribution:** 3
**Rating:** 4
**Confidence:** 3

**Summary:**

This research paper introduces a novel paradigm called Self-Improved Privilege Learning (SIPL) to address optimization instability and inter-task conflicts in all-in-one image restoration models when handling diverse and mixed degradations. Unlike conventional Privilege Learning, SIPL innovatively extends the utility of privileged information (PI) beyond the training phase into inference. Its core mechanism is the "Proxy Fusion" module, which incorporates a learnable Privileged Dictionary. During training, this dictionary distills high-quality priors from ground-truth features, and during inference, it leverages the model's preliminary outputs as pseudo-privileged signals for an iterative self-refinement loop. Experimental results demonstrate that SIPL significantly improves performance across various all-in-one image restoration benchmarks, particularly for composite degradation tasks, while offering broad applicability and computational efficiency.

**Strengths:**

1. SIPL breaks the limitations of traditional Privilege Learning by extending privileged information from the training phase to inference, enabling self-improvement at test time, which is a significant innovation.
2. Experimental results demonstrate that SIPL achieves substantial PSNR improvements across various image restoration tasks, including composite degradation, deraining, dehazing, and denoising, performing exceptionally well on complex composite degradations.
3. The SIPL framework, particularly the Proxy Fusion module, is designed to be seamlessly integrated with diverse backbone architectures (e.g., PromptIR, Restormer, NAFNet, AdaIR), enhancing its versatility and practicality.
4. The paper provides comprehensive ablation studies, deeply analyzing the contributions and performance of individual SIPL components, which enhances the credibility of the conclusions.

**Weaknesses:**

1. Deeper theoretical understanding needed: The paper points out a lack of a deeper theoretical understanding of the optimization dynamics of Privilege Learning in this context. While empirically validated for stability, the absence of a solid theoretical foundation might limit further optimization and insights.
2. Additional training cost and inference latency: This mthods needs retraining the baseline methods with the pluged module, and this training is needed for each new model. The iterative refinement process also increases latency compared to single-pass baselines.
3. Native privilege learning as the important baseline are somewhat missing. The main performance table only compares SIPL with the base model (PromptIR), ignoring the comparison with native privilege learning pipeline.
4. As the core component of this method, the learned dictionary lacks necessary analyses. Additional analyses are encouraged, regarding the impact of the size of the dictionary, and so on.

**Questions:**

Please refer to weakness part.

---

> ### Author Response · Authors · 2025-11-29
>
> We sincerely thank Reviewer n3ui for the positive assessment, highlighting the "significant innovation" of extending Privilege Learning to inference and recognizing our "substantial PSNR improvements" and "comprehensive ablation studies." We value your constructive feedback regarding the theoretical foundation and baseline comparisons. We have addressed these points with new analyses and experiments below.
>
> ---
>
> ### **Response to Weaknesses & Questions**
>
> **W1: Deeper theoretical understanding needed regarding optimization dynamics.**
>
> **Response:**
> We appreciate this suggestion to strengthen the paper's foundation. In the **Revised Appendix A**, we have added a comprehensive formal theoretical analysis that dissects the framework from three critical perspectives:
> 1.  **Motivation (Efficacy of Privilege Learning):** We analyze how the introduction of privileged information acts as a gradient rectifier, reducing the variance of stochastic gradients caused by random degradations and stabilizing the "tug-of-war" dynamics in multi-task optimization.
> 2.  **Methodology (Proxy Fusion):** We interpret the Proxy Fusion module as a **Manifold Projection** operator. The Privileged Dictionary (PD) spans a subspace of high-quality image features. During inference, the cross-attention mechanism projects noisy features onto this "clean" subspace, mathematically ensuring that the restored features remain within the valid signal manifold.
> 3.  **Stability Analysis:** We provide a convergence analysis of the multi-step inference process. Since the PD is frozen and bounded, the projection operator is non-expansive, theoretically guaranteeing that the iterative refinement converges to a stable state rather than diverging.
>
> **W2: Additional training cost and inference latency.**
>
> **Response:**
> We acknowledge the additional costs but emphasize that SIPL offers a highly flexible and efficient trade-off between performance and computation. Specifically, the "Native PL" configuration (Iter-0) incurs **zero additional inference latency** while providing significant performance gains purely through improved optimization dynamics. When higher performance is required, the iterative refinement (Iter-1) introduces manageable latency but delivers State-of-the-Art results much more efficiently than simply scaling up the model architecture. As demonstrated in our new experiments on the SIDD benchmark (see Appendix B.6), **NAFNet-32 + SIPL** matches the performance of the much larger **NAFNet-64**, achieving comparable results with approximately **31% fewer FLOPs**. This confirms that SIPL is a computationally efficient strategy for achieving top-tier restoration quality.
>
> **W3: Native privilege learning as the important baseline are somewhat missing.**
>
> **Response:**
> We clarify that the "Native Privilege Learning" baseline is indeed explicitly included and extensively analyzed throughout the manuscript. As our core contribution involves extending vanilla Privilege Learning (PL) by retaining the Privileged Dictionary, the result of our **Iter-0** stage represents the standard PL baseline. We have provided detailed comparisons between the base model, Iter-0 (Native PL), and Iter-1 (SIPL Refinement) in **Section 4.1**, the **Ablation Studies**, and **Appendix Tables 12, 13, and 16**. These comparisons clearly demonstrate that introducing Privilege Learning (Iter-0) significantly improves the multi-degradation model's baseline capabilities. Furthermore, the subsequent iterative refinement (Iter-1) effectively bridges the gap between the degraded features and the privileged priors, yielding further substantial performance gains.
>
> **W4: The learned dictionary lacks necessary analyses (e.g., impact of the size of the dictionary).**
>
> **Response:**
> To address this, we have conducted a detailed ablation study on the Privileged Dictionary (PD) size ($N$) ranging from 32 to 1024, with results plotted in **Figure 9** and discussed in **Revised Appendix B.2**. Our analysis reveals that performance improves steadily as the dictionary size increases, eventually saturating between 512 and 1024. Considering the trade-off between performance gains and computational overhead, we adopted **$N=256$** as the default setting for this paper. Beyond size, we also performed a deep dive into the intrinsic characteristics of the learned PD. We added comprehensive visualizations and clustering analyses in **Appendix B.1**, which confirm that the PD learns physically meaningful structural prototypes (e.g., specific textures and edges) rather than random noise, validating its role as a robust guide for restoration.

---

### Official Review · Reviewer_zHB3 · 2025-10-31

**Soundness:** 3
**Presentation:** 2
**Contribution:** 2
**Rating:** 4
**Confidence:** 4

**Summary:**

This paper introduces Self-Improved Privilege Learning (SIPL), a novel framework for all-in-one image restoration that extends privilege learning (PL) into the inference stage. The key idea is to enable models to iteratively refine their outputs by using their own initial restorations as pseudo-privileged information. The authors propose a Proxy Fusion module with a learnable Privileged Dictionary (PD) to retain high-quality priors from privileged (ground-truth-derived) features during training and reuse them during inference. The method is claimed to be architecture-agnostic and can be integrated into various backbones like PromptIR. Extensive experiments across multiple benchmarks (three-task, five-task, deweathering, and composite degradation) show notable PSNR/SSIM gains and strong qualitative improvements.

**Strengths:**

1. The paper presents a creative extension of Privilege Learning by introducing an inference-time reuse mechanism. The idea of “self-refinement through pseudo-privileged signals” is conceptually elegant and distinct from test-time adaptation or self-ensembling.
2. The proposed Proxy Fusion and Privileged Dictionary are well-motivated and described with clear mathematical formulation (Eqs. 2–4). The training and inference procedures are systematically explained, and the iterative refinement mechanism (Eqs. 5–7) is logically sound.
3. Experiments cover diverse benchmarks with consistent improvements over strong baselines.

**Weaknesses:**

1. The framework lacks formal analysis of why the Privileged Dictionary enables stable self-refinement. The paper acknowledges this as a limitation but providing any theoretical intuition (e.g., gradient variance reduction) would strengthen the work.
2. While the paper differentiates SIPL from test-time adaptation, the boundary remains somewhat blurred. A more rigorous comparison (quantitative or procedural) to self-distillation or self-training methods could better situate SIPL in the broader landscape.
3. Although the overhead is smaller than ensembling, repeated refinement still doubles inference time. The practical trade-off between latency and improvement could be more thoroughly quantified (e.g., FLOPs, runtime).
4. All benchmarks are synthetic; it remains unclear whether pseudo-privileged refinement helps under real-world degradations (e.g., RAW noise, ISP artifacts).
5. The paper treats the PD as a black box; no visualization of learned atoms or similarity between retrieved priors and ground truth is shown.

**Questions:**

1. What do PD entries represent visually or statistically? Can you visualize top-activated atoms or measure entropy / usage distribution?
2. How many refinement iterations are typically needed before saturation? Does performance ever degrade with more steps? Any evidence of oscillation?
3. If a PD is trained with one backbone (PromptIR), can it accelerate or improve another (Restormer) without retraining? This would demonstrate universality.
4. Have you tested SIPL on real-capture datasets (e.g., RainDS, LOL-V2, AIM-RealRain)? Does pseudo-privilege refinement still improve perceptual metrics (LPIPS/MUSIQ)?

---

> ### Author Response · Authors · 2025-11-29
>
> We sincerely thank Reviewer zHB3 for the encouraging feedback, describing our idea as "conceptually elegant," the formulation as "clear," and the results as "strong." We value your constructive critique regarding the theoretical analysis, inference dynamics, and real-world validation. We have addressed all points with additional experiments and comprehensive analyses below.
>
> ---
>
> ### **Part 1: Response to Questions**
>
> **Q1: What do PD entries represent visually or statistically? Can you visualize top-activated atoms or measure entropy/usage distribution?**
>
> **Response:**
> Thank you for this suggestion. Since the Privileged Dictionary (PD) operates in high-dimensional latent space, we have added a dedicated section in the **Revised Appendix (Sec. B.1)** to analyze it from two perspectives:
>
> 1.  **Clustering and Semantic Mapping:** We performed clustering analysis on the learned PD atoms. The results show tight, distinct clusters (visualized via t-SNE in the Appendix), indicating that the PD learns specific "prototypes" rather than random noise. By mapping top-activated atoms back to image space, we observed strong physical correlations: specific atoms consistently activate for distinct local structures (e.g., high-frequency textures, sharp edges, flat regions), confirming they serve as a structural codebook.
> 2.  **Feature Evolution Analysis:** We visualized the feature map differences between the initial pass (Iter-0) and the refinement pass (Iter-1).
>     * *Iter-0:* The PD-enhanced features show significantly stronger response to degradation patterns (e.g., rain streaks) compared to the baseline PromptIR.
>     * *Iter-1:* Using the pseudo-privileged output, the attention shifts to focus on *residual artifacts* and broken regions, validating our hypothesis that the PD facilitates a coarse-to-fine restoration.
>
> **Q2: How many refinement iterations are typically needed before saturation? Does performance ever degrade with more steps?**
>
> **Response:**
> We previously analyzed the 10-step iteration process in the initial manuscript (Table 6) and have now expanded this with more detailed metrics in the **Revised Appendix (Tables 22 & 23)**. As observed, the most significant gain occurs at the **first iteration**, after which performance rapidly saturates and stabilizes on a plateau. Crucially, due to our feature-level refinement via the fixed PD, the process avoids the error accumulation typical of naive pixel-level feedback, preventing performance degradation even after extended steps. Furthermore, we have added a **Theoretical Analysis** section in **Appendix A** to formally discuss the convergence properties and stability of our iterative mechanism.
>
> **Q3: If a PD is trained with one backbone (PromptIR), can it accelerate or improve another (Restormer) without retraining?**
>
> **Response:**
> Thank you for this insightful question regarding transferability. We investigated this by attempting to directly transfer a PD trained on PromptIR to a pre-trained Restormer. As shown in the table below, direct weight transfer leads to a significant performance drop due to the misalignment of latent feature distributions between different architectures.
>
> | Setting | Restormer (Base) | +SIPL (Joint Train) | +SIPL (Direct Transfer) | +SIPL (Transfer + Fine-tune) |
> | :--- | :---: | :---: | :---: | :---: |
> | **PSNR (dB)** | 27.60 | 28.57 | 11.33 | 28.71 |
>
> However, our *approach* is plug-and-play to existing models. As evaluated in experiments, training SIPL from scratch with diverse backbones (PromptIR, Restormer, NAFNet) consistently yields gains. Interestingly, using a pre-trained PD as initialization for fine-tuning on a new backbone (Restormer) can achieve slightly better results than training from scratch (28.71 dB vs 28.57 dB), suggesting the learned priors have some degree of generalizability when properly adapted.
>
> **Q4: Have you tested SIPL on real-capture datasets (e.g., RainDS, LOL-V2)? Does pseudo-privilege refinement still improve perceptual metrics?**
>
> **Response:**
> Yes, we have significantly expanded our real-world evaluation in the revised manuscript. In addition to the SIDD results provided previously, we added three new benchmarks covering diverse tasks:
> 1.  **Real-World All-in-One (WeatherBench):** SIPL achieves SOTA results, notably improving Dehazing by **+3.37 dB** and Desnowing by **+2.09 dB** (detailed discussion in **Sec. 4.1**).
> 2.  **Real-World Deraining (SPA):** Integrated with DRSformer and AST, SIPL consistently boosts performance (e.g., AST+SIPL reaches **49.92 dB** vs. 49.51 dB).
> 3.  **Real-World Low-Light (LOL-v2 Real):** Retinexformer+SIPL achieves **23.21 dB** (vs. 22.80 dB) on this challenging dataset.
>
> Detailed tables for these experiments are included in the **Revised Appendix (Sec. B.3, Tables 14 & 15)**.
>
> ---

---

> > ### Author Response · Authors · 2025-11-29
> >
> > ### **Part 2: Response to Weaknesses**
> >
> > **W1: The framework lacks formal analysis of why the Privileged Dictionary enables stable self-refinement.**
> >
> > **Response:**
> > We have added a comprehensive **Theoretical Analysis** in **Appendix A**, examining the effectiveness of Privilege Learning, the theoretical motivation behind our design, and the stability of the iterative convergence. Furthermore, to empirically validate these theoretical insights, we provide extensive visualization and analysis of the Privileged Dictionary's behavior in **Appendix B.1**.
> >
> > **W2: The boundary between SIPL and Test-Time Adaptation (TTA) remains somewhat blurred.**
> >
> > **Response:**
> > We clarified this distinction based on the definition from a recent survey (Liang et al., IJCV 2025). TTA relies on *gradient-based updates* to model parameters using *test data* to handle domain shifts. In contrast, SIPL employs a **frozen** model and dictionary, utilizing *inference-only refinement* based on priors distilled *offline* from Ground Truth. As shown in **Table B**, while SIPL outperforms standard TTA (+0.58 dB), they are complementary: applying TTA to fine-tune the SIPL Dictionary yields further gains (+0.07 dB). Detailed discussion and results are available in **Appendix B.7 (Tables 20 & 21)**.
> >
> > **Table B: SIPL vs. TTA on Three-Task Benchmark**
> > | Method | Mechanism | Cost | Avg PSNR |
> > | :--- | :--- | :--- | :--- |
> > | Baseline | One-pass | $1\times$ | 32.06 dB |
> > | Standard TTA | Gradient Update | $\sim 3000\times$ | 32.09 dB |
> > | **SIPL (Ours)** | **Feature Refinement** | **$2\times$** | 32.67 dB |
> > | SIPL + TTA | Refinement + Update | $\sim 3000\times$ | **32.74 dB** |
> >
> > **W3: Repeated refinement doubles inference time. The practical trade-off could be more thoroughly quantified.**
> >
> > **Response:**
> > We discuss the performance-computation trade-off extensively in **Appendix B.6**. SIPL offers a flexible spectrum: the training-only Privilege Learning provides a zero-cost boost, while iterative refinement allows for scalable gains. As evidenced by our comparison on SIDD, **NAFNet-32 + SIPL** (40.19 dB) nearly matches the performance of the much larger **NAFNet-64** (40.30 dB) while requiring **31% fewer FLOPs** (43.93G vs. 63.28G). This demonstrates that SIPL is a highly efficient strategy for achieving top-tier results compared to simply scaling up model architecture.
> >
> > **W4: Benchmarks are synthetic; unclear if it helps under real-world degradations.**
> >
> > **Response:**
> > In the original manuscript, we validated our method on real-world raw noise using the SIDD benchmark. In this revision, we have further extended this validation to real-world all-in-one tasks (WeatherBench), real rain (SPA), and real low-light conditions (LOL-v2). Please refer to **Response to Q4** for details.
> >
> > **W5: The paper treats the PD as a black box; no visualization of learned atoms is shown.**
> >
> > **Response:**
> > In the revised manuscript, we have included a detailed analysis of the Privileged Dictionary (PD), comprising theoretical justification for its effectiveness and comprehensive visualizations of the learned atoms. Please refer to **Response to Q1** and **Appendix B.1** for these additions.

---

### Official Review · Reviewer_T5Qh · 2025-10-31

**Soundness:** 3
**Presentation:** 2
**Contribution:** 3
**Rating:** 6
**Confidence:** 3

**Summary:**

This paper proposes a new image restoration framework, Self-Improved Privilege Learning (SIPL), designed to address optimization instability and inter-task conflicts in all-in-one image restoration models. Built upon the concept of Privilege Learning (PL), the authors extend its use beyond training to inference via a lightweight module called Proxy Fusion, which incorporates a learnable Privileged Dictionary (PD). At inference, the model uses its own outputs as pseudo-privileged information, enabling iterative self-refinement. The method is shown to be architecture-agnostic, efficient, and broadly applicable, with strong results on multiple benchmarks, including multi-task, composite degradation, and out-of-distribution scenarios.

**Strengths:**

1. The extension of PL into test-time self-refinement using a learned dictionary is both conceptually interesting and practically effective.
2. The proposed Proxy Fusion mechanism is lightweight, plug-and-play, and well-motivated. It introduces minimal overhead while providing measurable improvements.
3. Comprehensive experiments across four challenging benchmarks (Three-Task, Five-Task, Deweathering, Composite Degradation) convincingly demonstrate SIPL’s effectiveness. Improvements are consistently reported across various restoration tasks, with particularly notable gains in composite degradation scenarios.
4. The paper includes detailed ablations dissecting PL and SIPL contributions, multi-step refinement behavior, and efficiency vs. performance trade-offs. These analyses are thorough and help clarify SIPL’s practical value.

**Weaknesses:**

1. The paper is difficult to read in many parts due to heavy terminology and dense writing.
2. While the empirical results are strong, the paper lacks a deeper theoretical analysis of why the proposed self-refinement via pseudo-PI is stable and effective. A formal justification or insight into training dynamics under SIPL would strengthen the claims.
3. The performance of SIPL at inference appears to depend heavily on the quality of the initial model output. If the initial restoration is poor, the pseudo-privileged signal may be too noisy to guide useful correction. This limitation is acknowledged but not explored further.
4. The data for Denoising in Table 2 appears to be incorrect. 31.45 is incorrectly bolded, but SSIM is indeed higher.

**Questions:**

1. How does the performance of SIPL degrade if the initial restoration is poor? Are there failure cases?
2. Is there any benefit to fine-tuning the Privileged Dictionary during inference, or is it always fixed?
3. What the performance of SIPL under real-world degradations not covered by the benchmark datasets?

---

> ### Author Response · Authors · 2025-11-29
>
> We sincerely thank Reviewer T5Qh for the insightful feedback and positive assessment, particularly regarding the novelty of extending Privilege Learning to the inference stage and the effectiveness of the Proxy Fusion mechanism. We have carefully revised the manuscript to address your concerns, adding new real-world benchmarks, theoretical analysis, and clarification on failure modes.
>
> Below are our detailed responses.
>
> ---
>
> ### **Part 1: Response to Specific Questions**
>
> **Q1: How does the performance of SIPL degrade if the initial restoration is poor? Are there failure cases?**
>
> **Response:**
> We appreciate this consideration regarding robustness. We analyzed this from two perspectives: generalization on unseen data (where initial restoration is suboptimal) and stability in severe failure scenarios.
>
> 1.  **Generalization to Unseen Degradations (CDD-11):**
>     To test performance when the initial restoration is likely poor, we evaluated our model (trained *only* on the 5-task setting) directly on the **CDD-11 dataset** containing unseen composite degradations (e.g., Haze+Rain). SIPL consistently outperforms the baseline PromptIR across all 11 unseen degradation types. This confirms that the Privileged Dictionary (PD) successfully guides refinement even when the input distribution shifts.
>
> 2.  **Stability in Failure Cases:**
>     We stress-tested the model on real-world heavy snow images involving massive occlusion (visualized in the **Revised Appendix**). While the model cannot hallucinate missing content behind heavy snow, the iterative process remains **stable**. Unlike pixel-level feedback loops that often hallucinate artifacts when the input is poor, our feature-level refinement acts conservatively. The Proxy Fusion module projects features onto the "clean" manifold spanned by the PD, preventing error amplification.
>
> Detailed discussion and visualization results are added in **Revised Appendix (Sec B.5)**.
>
> **Q2: Is there any benefit to fine-tuning the Privileged Dictionary during inference, or is it always fixed?**
>
> **Response:**
> For the main results, the Dictionary is **fixed**. However, to address this interesting question, we explored a **Test-Time Adaptation (TTA)** setting where we fine-tune the PD on the test image using a self-supervised loss.
>
> As shown in **Table B**, standard TTA (tuning the encoder) yields negligible gains because the test domain is close to the training domain. However, **SIPL + TTA** (tuning only the PD) yields noticeable improvements (e.g., +0.23 dB in Deraining). This suggests that while a fixed PD is highly effective, fine-tuning can further bridge the domain gap between the stored priors and the specific test instance. We have added this discussion to the **Appendix**.
>
> **Table B: Adaptation Strategies on Three-Task Benchmark**
> | Strategy | Deraining | Dehazing | Denoise (Avg) | **Average** |
> | :--- | :---: | :---: | :---: | :---: |
> | Baseline (PromptIR) | 36.37 | 30.58 | 31.12 | 32.06 |
> | Standard TTA (Full Model) | 36.43 | 30.71 | 31.10 | 32.09 |
> | **SIPL (Fixed PD - Default)** | **38.43** | **31.09** | **31.27** | **32.67** |
> | SIPL + TTA (Tune PD) | 38.66 | 31.28 | 31.26 | **32.74** |
>
> **Q3: What is the performance of SIPL under real-world degradations not covered by the benchmark datasets?**
>
> **Response:**
> To address the "synthetic-only" concern, we evaluated our method on the WeatherBench dataset, a large-scale real-world benchmark. As detailed in the Table 4 of **Revised Manuscript** , SIPL achieves State-of-the-Art performance, significantly outperforming the baseline PromptIR, particularly in Dehazing (**+3.37 dB**) and Desnowing (**+2.09 dB**). This confirms that the structural priors captured by the PD are degradation-agnostic and transfer effectively to complex real-world environments.
>
> ---

---

> > ### Author Response · Authors · 2025-11-29
> >
> > ### **Part 2: Response to Weaknesses**
> >
> > **W1: The paper is difficult to read due to heavy terminology.**
> >
> > **Response:**
> > We sincerely apologize for the dense writing style and the confusion caused by heavy terminology. We will simplify the definitions of core concepts and streamline the presentation of the technical workflow to ensure the paper is clearer and easier to follow.
> >
> > **W2: Lack of deeper theoretical analysis on stability and effectiveness.**
> >
> > **Response:**
> > Thank you for this crucial suggestion. We have added a **Theoretical Analysis** section in the **Revised Appendix A**, analyzing SIPL through the lens of **Manifold Projection**:
> > 1.  **Clean Manifold Constraint:** The Privileged Dictionary acts as a set of basis vectors ($\Phi$) spanning the subspace of high-quality image features. Since the dictionary is frozen at inference, these atoms remain strictly within the clean manifold.
> > 2.  **Projection Stability:** The Proxy Fusion module mathematically functions as a projection operator. Even if the initial feature query is noisy, the output is constructed as a linear combination of these clean atoms. This structural constraint ensures the output remains "plausible," preventing the error amplification typically seen in naive feedback loops and ensuring rapid convergence (as empirically verified in our convergence experiments).
> >
> > **W3: Dependence on initial model output quality.**
> >
> > **Response:**
> > Please refer to our response to **Q1**. Our additional experiments on OOD data (CDD-11) and failure case analysis confirm that the method is robust. The **Manifold Projection** mechanism (discussed in W2) ensures that even poor initial inputs are projected to the nearest valid clean feature representation rather than diverging.
> >
> > **W4: Representation error in Table 2 (Denoising bolding).**
> >
> > **Response:**
> > We have corrected this style formatting in the revised manuscript. Thank you for your meticulous review.

---

### Author Response · Authors · 2025-11-29

**To All Reviewers:**

We sincerely thank you for your time and the constructive feedback provided during the review process. Your insights have been invaluable in refining the quality and rigor of our work. We have carefully considered all comments and have updated the manuscript accordingly.

We would like to highlight the **four major updates** in this revision:

**1. Comprehensive Theoretical Analysis (Appendix Sec. A)**
In response to requests for a deeper theoretical grounding, we have added a formal analysis section that dissects the SIPL framework from three perspectives:
* **Motivation:** We analyze the efficacy of Privilege Learning as a form of gradient rectification that stabilizes multi-task optimization.
* **Methodology:** We interpret the Proxy Fusion module through the lens of **Manifold Projection**, explaining how it constrains features to a high-quality subspace.
* **Stability:** We provide a convergence analysis of the iterative inference process, theoretically justifying why the self-refinement remains stable and avoids divergence.

**2. Extensive Real-World Validation (Sec. 4.1 & Appendix Sec. B.4)**
To address concerns regarding synthetic benchmarks, we have significantly expanded our experimental evaluation to include complex real-world scenarios:
* **All-in-One:** We achieve State-of-the-Art results on the large-scale **WeatherBench** dataset.
* **Single Tasks:** We validated our method on real-world deraining (**SPA** dataset) and low-light enhancement (**LOL-v2 Real** dataset).
These results confirm the robustness of our method beyond synthetic data.

**3. In-depth Ablation Studies & Comparisons (Sec. 4.2 & Appendix Sec. B.1, B.2, B.8)**
We have conducted a more granular analysis of our core components:
* **PD Characteristics:** We visualize the clustering properties and physical meaning of the Privileged Dictionary atoms (Sec. 4.2, Appendix B.1).
* **Hyperparameters:** We analyze the impact of dictionary size on performance and efficiency (Appendix B.2).
* **TTA Comparison:** We provide a rigorous comparison with **Test-Time Adaptation (TTA)**, demonstrating the distinct mechanisms and complementary benefits of our approach (Appendix B.8).

**4. Generalization and Robustness Analysis (Appendix Sec. B.6)**
We have performed a thorough stress-test of our framework:
* **Cross-Domain Validation:** We evaluate the model on unseen composite degradation types (e.g., assessing a 3-task model on the CDD-11 dataset).
* **Failure Case Analysis:** We transparently discuss limitations by analyzing performance under severe degradations (e.g., heavy snow), demonstrating the stability of our feature-level refinement even in challenging failure modes.

We believe these revisions comprehensively address the reviewers' concerns and significantly strengthen the contributions of our paper. We invite you to review the revised manuscript and appendices.

Best regards,

The Authors

---

### Meta-Review · Area_Chair_tzFq · 2025-12-23

**Summary:**

After the rebuttal, this paper recevices two marginally below the acceptance threshold and one marginally above the acceptance threshold. There still exist several major issues with this paper. (1) There is no comparison of the running time and flops with existing methods. As the proposed method is in an iterative manner, the analysis of the computational cost is important. (2) Even though discussed in the rebuttal, the theoretical analysis of the privileged dictionary and self-distillation is still not satisfactory. As a result, the paper cannot be accepted in the current form.

**Reviewer Scores:**

NA

---

### Decision · Program_Chairs · 2026-01-26

Reject